# Identifying potential biomarkers and therapeutic targets for dogs with sepsis using metabolomics and lipidomics analyses

Brett Montague[ID][1☯], April Summers[1☯], Ruchika Bhawal[2], Elizabeth T. Anderson[2], Sydney Kraus-Malett[1], Sheng Zhang[2], Robert Goggs[ID][1]*

1 Department of Clinical Sciences, College of Veterinary Medicine, Cornell University, Ithaca, New York, United States of America, 2 Proteomics and Metabolomics Facility, Cornell University, Ithaca, New York, United States of America

☯ These authors contributed equally to this work.

* r.goggs@cornell.edu

**Data Availability Statement:** All relevant data are within the manuscript and its Supporting Information files.

## Abstract

Sepsis is a diagnostic and therapeutic challenge and is associated with morbidity and a high risk of death. Metabolomic and lipidomic profiling in sepsis can identify alterations in metabolism and might provide useful insights into the dysregulated host response to infection, but investigations in dogs are limited. We aimed to use untargeted metabolomics and lipidomics to characterize metabolic pathways in dogs with sepsis to identify therapeutic targets and potential diagnostic and prognostic biomarkers. In this prospective observational cohort study, we examined the plasma metabolomes and lipidomes of 20 healthy control dogs and compared them with those of 21 client-owned dogs with sepsis. Patient data including signalment, physical exam findings, clinicopathologic data and clinical outcome were recorded. Metabolites were identified using an untargeted mass spectrometry approach and pathway analysis identified multiple enriched metabolic pathways including pyruvaldehyde degradation; ketone body metabolism; the glucose-alanine cycle; vitamin-K metabolism; arginine and betaine metabolism; the biosynthesis of various amino acid classes including the aromatic amino acids; branched chain amino acids; and metabolism of glutamine/glutamate and the glycerophospholipid phosphatidylethanolamine. Metabolites were identified with high discriminant abilities between groups which could serve as potential biomarkers of sepsis including 13,14-Dihydro-15-keto Prostaglandin A$_2$; 12(13)-DiHOME (12,13-dihydroxy-9Z-octadecenoic acid); and 9-HpODE (9-Hydroxyoctadecadienoic acid). Metabolites with higher abundance in samples from nonsurvivors than survivors included 3-(2-hydroxyethyl) indole, indoxyl sulfate and xanthurenic acid. Untargeted lipidomic profiling revealed multiple sphingomyelin species (SM(d34:0)+H; SM(d36:0)+H; SM(d34:0)+HCOO; and SM (d34:1D3)+HCOO); lysophosphatidylcholine molecules (LPC(18:2)+H) and lipophosphoserine molecules (LPS(20:4)+H) that were discriminating for dogs with sepsis. These biomarkers could aid in the diagnosis of dogs with sepsis, provide prognostic information, or act as potential therapeutic targets.

**Funding:** This study was funded by a Nestlé Purina Resident's Research Grant to AS and RG. The funders had no role in study design, data collection and analysis, decision to publish, or preparation of the manuscript.

**Competing interests:** The authors have declared that no competing interests exist.

## Introduction

Sepsis is a consequence of a dysregulated host response to infection resulting in organ damage [1], characterized by circulatory, cellular, and metabolic derangements that are life-threatening, with case fatality rates reported to be up to 70% in dogs [2]. Humans with sepsis exhibit a catabolic state with rapid breakdown of protein, carbohydrate, and fat reserves causing severe energy deficits [3]. Moreover, many humans also have gastrointestinal dysfunction that increases the risk of malnutrition [4]. The metabolic derangements in sepsis are myriad [5], and impair wound healing, reduce gut function, and enable intestinal bacterial translocation [6]. Nutritional treatments can improve clinical outcomes by attenuating the metabolic response to stress, reducing oxidative injury, and modulating the immune response [6]. Trials in humans with sepsis have shown that targeted nutritional support reduces organ dysfunction, hospital-acquired infections, and death [7, 8] and retrospective studies of dogs with sepsis suggest that early enteral nutrition might decrease length of hospitalization [9], and improve survival rates [10].

Metabolomics utilizes high-performance chromatography and tandem mass spectrometry to simultaneously identify, quantify, and characterize large numbers of metabolites in complex biological samples [11]. However, unambiguous identification of metabolites in complex matrices remains challenging due to the lack of appropriate public metabolome databases. This has led to the emergence of new comprehensive approaches to improve accuracy and confidence in metabolite identification. Comprehensive metabolic profiling offers huge potential for pathophysiologic insight, biomarker discovery, and identification of therapeutic targets [12]. These techniques have been applied to humans with sepsis [13], and have identified some specific derangements in lipid metabolism and amino acid handling [14], but similar studies have not been performed in dogs. Lipidomics is a branch of metabolomics that specifically identifies and quantifies the lipid molecules of major lipid classes to aid in the understanding of cellular lipid metabolic pathways. Technological improvements in both chromatography and mass spectrometry instruments and methods aid the confident identification and quantitation of lipid molecules and have helped establish the importance of lipids in sepsis. Lipidomics is facilitating improvements in our understanding of sepsis pathogenesis and the identification of new biomarkers [15]. Studies in mice indicate that numerous plasma lipid species are quantitatively altered in sepsis [16]. The administration of lipid nutritional supplements ameliorates some of these effects and reduces case fatality rates [17]. Various mechanisms have been proposed for these alterations in lipid profiles. Eicosanoids, $\Omega$-3 and $\Omega$-6 fatty acids, lysophosphatidylcholine, and ceramide play significant roles in sepsis [18] and plasma lysophosphatidylcholine, sphingomyelin and unsaturated phosphatidylcholine concentrations are altered in humans with sepsis [19]. It is also known that pro-inflammatory cytokines significantly impact lipid metabolism, alter lipoprotein metabolism, and decrease total cholesterol concentrations [20].

The overall objectives of our study were to characterize the metabolic derangements in dogs with sepsis and to identify potential diagnostic and prognostic biomarkers in their metabolomes. The study aimed to use global untargeted metabolomics and lipidomics approaches to discover novel metabolite biomarkers [21], enhance our understanding of the metabolic derangements in sepsis and suggest pathways that might be amenable to therapeutic interventions. It was hypothesized that dogs with sepsis have metabolomic and lipidomic profiles that are distinct from those of healthy control dogs; that significantly changed levels of individual metabolites in dogs with sepsis compared to healthy controls would allow us to identify putative biomarkers and or their patterns for early diagnosis of the syndrome; and that the presence of particular metabolites or metabolic profiles would be prognostic for survival in dogs with sepsis.

## Materials and methods

### Study design

This was a prospective observational cohort study of client-owned dogs with sepsis admitted to the Cornell University Hospital for Animals. Dogs were eligible for enrollment if they weighed >5kg, had a documented or highly suspected infection (such as pyometra, septic peritonitis or pneumonia) and satisfied ≥2 systemic inflammatory response syndrome (SIRS) criteria, specifically, hypo- or hyperthermia temperature <37.8°C or >39.4°C (<100.0°F or >102.9°F); tachycardia, heart rate >140 bpm; tachypnea, respiratory rate >20 bpm; leukopenia or leukocytosis, <6 ×$10^3$/μL or >16 ×$10^3$/μL or >3% band neutrophils [22, 23].

Dogs were ineligible if they had sepsis due to viral disease e.g., parvovirus or fungal disease e.g., candidiasis, severe anemia, coagulopathy, or thrombocytopenia (Hb <5 g/dL; PT or aPTT >150% normal; platelets <30 ×$10^3$/μL). Dogs not expected to live more than 12 hours or those with pre-existing metabolic conditions or endocrinopathies such as diabetes mellitus or hypothyroidism were also excluded. Dogs were enrolled with written informed client consent. The local Institutional Animal Care and Use Committee approved the study protocol (Cornell IACUC Protocol #2014–0053). There are no standardized methods for sample size estimation in metabolomics studies, and traditional sample size calculation approaches are not easily applied to untargeted metabolic phenotyping studies [24]. Preliminary data (unpublished) suggested that 20 dogs per group would be sufficient to distinguish sepsis cases from controls and data from a comparable study in humans [25] suggested this number of animals would be sufficient to identify potential prognostic markers. Healthy dogs were recruited from staff-owned pets and were eligible for the study if they weighed >5kg, were aged between 1-9y, had no chronic or recent illness, and had received no medications other than preventative healthcare (e.g., parasiticides) in the preceding three months. Dogs were classified as healthy based on history, physical examination, and the results of complete blood count and serum biochemistry profile results.

### Case management and evaluation

Respective primary clinicians determined all aspects of case management. Signalment and physical examination findings at hospital admission were recorded. Blood gases, electrolytes and lactate concentrations were measured immediately after sample collection with a point-of-care device (RapidPoint 500, Siemens Healthcare, Malvern, PA). Complete blood counts (CBC) (ADVIA 2120, Siemens Healthcare) with clinical pathologist review and serum biochemistry profiles (Cobas C501, Roche Diagnostics, Indianapolis, IN) were analyzed immediately whenever possible, but always within 48 hours of collection. Mentation score, blood glucose, albumin and lactate concentrations and platelet counts were used to calculate the acute patient physiologic and laboratory evaluation illness severity score (APPLE$_{fast}$) [26, 27]. Outcome status at discharge was recorded as survived, died or euthanized. Blood samples were collected at study entry into evacuated tubes (Vacutainer, BD and Co, Franklin Lakes, NJ) containing no-additive (for serum biochemistry analyses), lithium heparin (for metabolomics and lipidomics) or K$_2$-EDTA (for complete blood counts). Heparin plasma was prepared from whole blood by centrifugation for 10 minutes at 1370 g (Ultra-8V Centrifuge, LW Scientific, Lawrenceville, GA). Plasma was decanted into polypropylene freezer tubes (Polypropylene Screw-Cap Microcentrifuge Tubes, VWR, Radnor, PA) with some plasma deliberately left in each tube to minimize the risk of cell contamination and frozen at -80°C pending batch analysis.

## Untargeted metabolomics

Plasma samples obtained from 20 healthy dogs and 21 dogs with sepsis were thawed at 4˚C, gently vortexed and 200 μL of each sample were transferred into 1.5 mL microcentrifuge tubes and stored on ice (Fig 1). The volume for one sepsis dog sample (#18) was very limited, for this sample only 100 μL was analyzed. To each sample, 600 μL of ice cold 100% methanol was added, the samples vortexed for 10 s and then incubated at 4˚C for 60 min for protein precipitation. After incubation, samples were centrifuged (16,200 g, 10 min, 4˚C) and 300 μL of the supernatants transferred into two clean 1.5 mL microcentrifuge tubes (one for each column type). The methanol volume for protein precipitation was halved for sample #18. The supernatants obtained were then evaporated to dryness by speed vacuum and stored at -20˚C for further reconstitution in buffer for LC MS/MS analysis. Each dried sample was reconstituted in 80 μL of 20% acetonitrile with 0.1% formic acid containing a panel of 3 internal standards

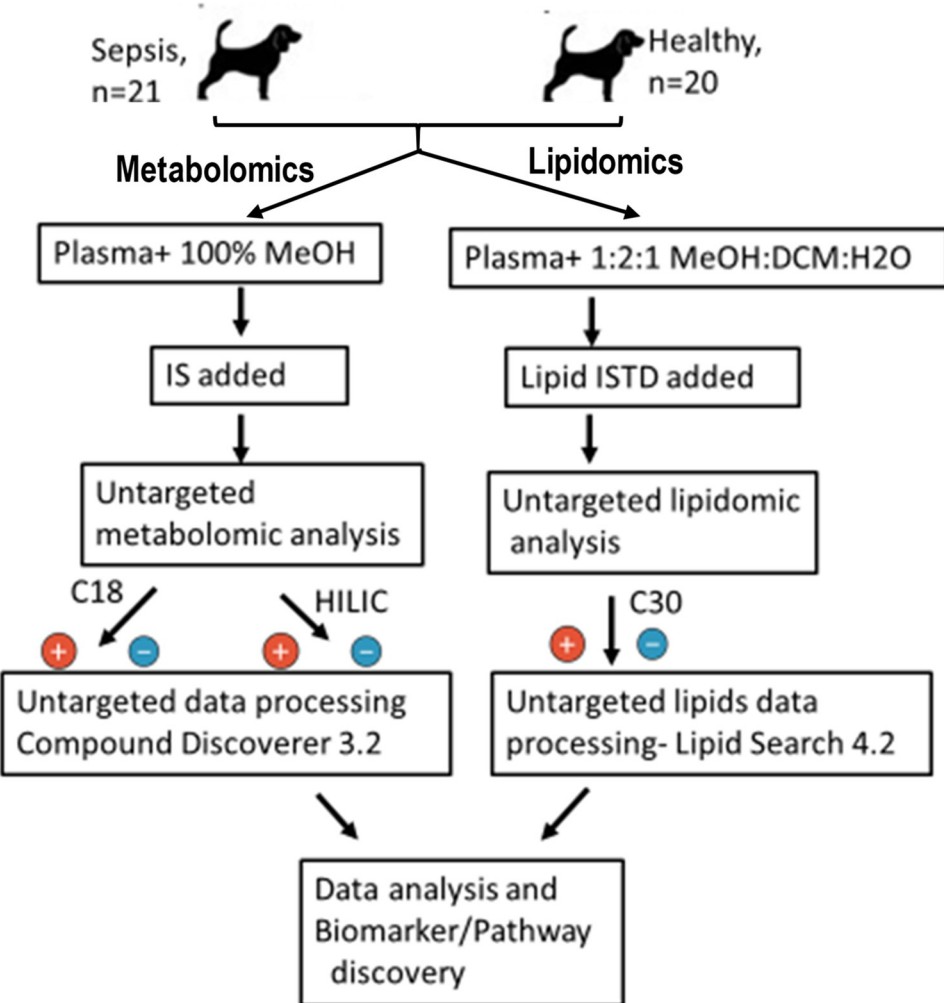

**Fig 1. Schematic diagram for untargeted metabolomics and lipidomics workflow for analysis of plasma samples collected from dogs with sepsis and from healthy controls.** Small-molecule metabolites and lipids were extracted from the sample matrix. Metabolites and lipids were separated using chromatographic steps, ionized, and analyzed using mass spectrometry (MS). Features of interest were selected from the raw data using univariate and multivariate statistical approaches and then identified using database searches, comparisons to authentic standards, and MS/MS. Identified features were then used for pathway analysis, and to distinguish potential metabolite biomarkers.

-0.5 ppm sulfadimethoxine (SFDT), 2.5 ppm [13]C pyruvic acid and 2.5 ppm [13]C valine. The reconstituted samples were then analyzed in LC MS/MS using reverse phase octadecyl-carbon chain (C18) column for identification of non-polar metabolites. For hydrophilic interaction chromatography (HILIC) analysis, each dried sample was reconstituted in 80 μL of 20% aceto-nitrile containing 0.5 ppm SFDT, 2.5 ppm [13]C pyruvic acid and 2.5 ppm [13]C valine. Added volumes were halved for sample #18. For all metabolomic analyses, the following quality controls (QC) were made: one global QC consisted of a pool of 3 μL from all samples (n = 41, ∑ 123 μL); two group QCs with one control QC consisting of a pool of 3 μL from all healthy control samples (n = 20, ∑ 60 μL); and another sepsis QC containing a pool of 3 μL from all sepsis samples (n = 20, ∑ 60 μL).

For C18 analysis, a Vanquish UPHLC system with a 1.5 μm column (2.1 mm id x 100mm) (Accucore Vanquish C18+, Thermo Fisher Scientific, Waltham, MA) at 45˚C was used for the separation of metabolites. Solvent flow rate was 320 μL/min. The autosampler tray was held at 4˚C and sample injection volume was 2 μL. The following solvents and elution gradients were used. Solvent A: Water / 0.1% formic acid; Solvent B: Acetonitrile / 0.1% formic acid. Elution gradient: 0.0–2.0 min (0.5–1% B), 2.0–6.5 min (1–20% B), 6.5–11.5 min (20–95% B), 11.5–13.5 min (95–99% B), 13.5–16.5 min (99–100% B), 16.5–19 min (100–0.5% B), 19–24 min (0.5% B). In the case of HILIC analysis for the identification of polar metabolites in plasma samples, a column (5μm, 2.1 mm id x 150mm) (SeQuant ZIC pHILIC, Millipore Sigma, Burlington, MA) at 24˚C was used. Solvent flow rate was 250 μL/min. The autosampler tray was held at 4˚C and sample injection volume was 2 μL. The following solvents and elution gradients were used. Solvent A: 10 mM AcONH$_4$ in H$_2$O, pH 9.8; Solvent B: Acetonitrile. Elution gradient: 0.0–1.0 min (90% B), 1.0–15 min (90–30% B), 15–18 min (30% B), 18–19 min (30–90% B), 19–29 min (90% B). The experimental conditions for the tandem mass spectrometry MS/MS were the same for both C18 and HILIC analyses. Tandem MS analyses were performed on an orbitrap mass spectrometer (Q-Exactive Hybrid Quadrupole-Orbitrap, Thermo Fisher Scientific, San Jose, CA). The ESI voltage was kept at 3.5 kV, the sheath gas flow rate was 50 AU, the auxiliary gas flow rate was 10 AU, and the sweep gas flow rate was 1 AU. The capillary temperature was 275˚C and the auxiliary gas heater temperature was 375˚C. The S-Lens RF level was 55% and all analyses were conducted in both positive and negative ion modes.

Data analysis was conducted using commercial software (Compound Discoverer 3.1, Thermo Fisher Scientific) for normalization, peak alignment, related statistical analyses and compound identification. An in-house mzVault spectral library and the public mzCloud database were used to annotate compounds on an MS/MS level with a mass tolerance of 10ppm and additional databases including ChemSpider, BioCyc, Human Metabolome Database, and KEGG database were searched for annotations and pathway analyses. The initially identified molecules in plasma samples were filtered out in CD3.1 through background subtraction and exclusion of false positive or repetitive features without MS2 spectra, and removal of compounds not found in QC samples. Separate software packages were used for creating supervised models including OPLS-DA (orthogonal partial least-squares discriminant analysis) for making score plots (Simca P, Umetrics, Sartorius, Goettingen, Germany) on filtered data and volcano plots, heatmap construction and pathway analysis (Metaboanalyst 5.0, https://www.metaboanalyst.ca, [28]). Univariate analysis based on ROC (receiver operating characteristic) curves was also performed to identify potential biomarkers for sepsis.

## Untargeted lipidomics

Plasma samples from two different groups of dogs (20 controls and 21 sepsis cases) were thawed at 4˚C, gently vortexed and 30 μL were transferred into high G-force 1.5 mL

microcentrifuge tubes (catalog #20170–038, VWR, Radnor, PA) and stored on ice. To each sample, 30 μL of a panel of 7 internal standards containing 25 μg/mL each of triglycerides $(15:0)_3$, phosphatidylglycerol $(14:0)_2$, phosphatidylserine $(16:0)_2$, ceramide (d18:1_12:0), cholesterol (17:0) and 5 μg/mL lysophosphatidylcholine (18:1-d7), phosphatidylcholine (18:1-d7_15:0) (Avanti Polar Lipids Inc, Alabaster, AL) in dichloromethane/methanol (2:1) were added for normalization of each lipid class (S1 Data). In addition, 190 μL of ice-cold 100% methanol, 380 μL of dichloromethane and 120 μL of water were added to each sample with vortexing. Sample mixtures were allowed to equilibrate for 10 minutes at room temperature, followed by centrifugation (18,000 g, 10 min, 4˚C). Gel loading tips were then used to transfer 350 μL (175 μL twice) of the lower lipid rich phase into a clean glass culture tube. Samples were then evaporated to dryness by speed vacuum, capped and stored at -20˚C for further lipid analysis in LC MS/MS. In preparation for lipid analysis, samples were reconstituted with 360 μL of acetonitrile / isopropanol / water (65:30:5 v/v).

Lipidomic analysis were carried out using a column (2.6 μm, 2.1 mm id x 150mm) (Accucore C30, Thermo Fisher Scientific) at 55˚C. Solvent flow rate was 260 μL/min. The autosampler tray was held at 4˚C and sample injection volume was 2 μL. The following solvents and elution gradients were used. Solvent A: 60% acetonitrile, 40% water, 10 mM ammonium formate with 0.1% formic acid. Solvent B: 90% isopropanol, 10% acetonitrile, 10 mM ammonium formate with 0.1% formic acid. Elution gradient: 0.0–1.5 min (32% B), 1.5–4.0 min (32–45% B), 4.0–5.0 min (45–52% B), 5.0–8.0 min (52–58% B), 8.0–11 min (58–66% B), 11–14 min (66–70% B), 14–18 min (70–75% B), 18–21 min (75–97% B), 21–25 min (97% B), 25–30 min (97–32% B). The experimental conditions for the tandem mass spectrometry for C30 analyses were as follows. The ESI voltage was 4 kV, the sheath gas flow rate was 50 AU, the auxiliary gas flow rate was 5 AU, and the sweep gas flow rate was 1 AU. The capillary temperature was 320˚C and the auxiliary gas heater temperature was 350˚C. The S-Lens RF level was 50% and all analyses were run in both positive and negative ion modes.

Data analysis was conducted using commercial software (LipidSearch 4.2, Thermo Fisher Scientific) to perform normalization, peak alignment, compound identification using the online Lipid Maps database and related statistical analyses. Other statistical analyses of these data were as described above for untargeted metabolomics analyses.

### General statistical methods

Continuous data (e.g., dog characteristics, physical examination findings and clinicopathologic values) were assessed for normality using the D'Agostino Pearson test and appropriate descriptive statistics calculated. Correlations between C18 and HILIC prognostic biomarker abundance and illness severity scores (APPLE$_{fast}$) were calculated using non-parametric methods (Spearman's r) following natural log transformation of the abundance data. General statistical analyses were performed using commercial software (Prism 9 for macOS, GraphPad, La Jolla, CA) with alpha set at 0.05.

## Results

### Animals

A total of 41 dogs were enrolled; 21 dogs with sepsis and 20 healthy controls. The 21 dogs with sepsis had a variety of different causes and mechanisms, specifically 4 dogs had abscesses or cellulitis, 3 dogs had peritonitis, 3 dogs had pneumonia, 3 dogs had pyometra, and 2 dogs had mastitis. Other causes included anaplasmosis, gastroenteritis (with bacteremia), metritis, osteomyelitis, pyothorax and urosepsis (all n = 1). Of the 21 dogs, 3 were euthanized for disease severity prior to discharge, the remainder survived to discharge, equivalent to a 14% case

fatality rate. Of the 18 dogs that survived to hospital discharge, 17 dogs were alive at day 28, with 1 dog lost to follow up, equivalent to a 15% 28-day case fatality rate. Demographic characteristics, initial assessments and clinicopathologic variables are summarized in Table 1. Dogs were prescribed a variety of medications prior to study enrolment, summarized in Table 2. Positive cultures were obtained in 67% (10/15) dogs for which culture samples were submitted. Various bacterial organisms were cultured from the dogs including Escherichia coli (n = 5), *Bacteroides* spp. (n = 2), *Clostridium perfringens* (n = 2), *Staphylococcus pseudintermedius* (n = 2), *Actinomyces canis*, *Enterococcus faecium*, *Fusobacterium* sp., *Microbacterium phyllosphaerae*, *Mycoplasma* sp., *Peptostreptococcus* sp., *Pseudarthrobacter* sp., and *Streptococcus canis* (all n = 1). The diets were known for 14/21 dogs with sepsis; all were eating commercial cooked diets from various manufacturers and 7 were eating more than one type of food. The

**Table 1. Summary of population characteristics including complete blood count and serum biochemistry data from study entry.**

| Variable (SI units) | Dogs with sepsis (n = 21) | Healthy control dogs (n = 20) |
|---|---|---|
| Age (y) | 4.4 ± 3.4 | 4.6 ± 2.6 |
| Bodyweight (kg) | 27.7 ± 14.9 | 33.9 ± 12.3 |
| Sex (F/FS/M/MC) | 6 / 6 / 3 / 5 | 0 / 13 / 1 / 6 |
| T (˚C) | 39.4 (38.3–40.6) | - |
| HR (bpm) | 142 ± 21 | - |
| RR (bpm) | 32 (26–40) | - |
| SAP (mmHg) | 139 ± 32 | - |
| MAP (mmHg) | 107 ± 27 | - |
| DAP (mmHg) | 92 ± 27 | - |
| $SpO_2$ (%) | 96 ± 3 | - |
| SIRS criteria (n) | 3 (3–3) [Max 4] | - |
| APPLE$_{fast}$ score | 22 (17–26) [Max 50] | - |
| LoH (d) | 3 (2–5) | - |
| AMD duration (d) | 16 (12–23) | - |
| AMDs prescribed (n) | 3 (3–4) | - |
| AMD classes (n) | 2 (2–3) | - |
| Lactate | 2.0 (1.4–3.4) | - |
| BG (mg/dL) | 97 (82–113) | 97 (90–103) |
| HCT (%) | 44 ± 8.4 [41–58] | 51 ± 6.3 [41–58] |
| Leukocytes (×10$^9$/L) | 16.6 ± 8.0 [5.7–14.2] | 8.4 ± 4.0 [5.7–14.2] |
| Neutrophils (×10$^9$/L) | 11.4 ± 8.0 [2.7–9.4] | 5.2 ± 3.2 [2.7–9.4] |
| Bands (×10$^9$/L) | 1.6 (0.4–3.5) [0.0–0.1] | 0.0 (0.0–0.0) [0.0–0.1] |
| Lymphocytes (×10$^9$/L) | 1.2 (0.5–2.9) [0.9–4.7] | 2.0 (1.3–2.3) [0.9–4.7] |
| Monocytes (×10$^9$/L) | 1.1 (0.5–2.0) [0.1–1.3] | 0.3 (0.3–0.5) [0.1–1.3] |
| Eosinophils (×10$^9$/L) | 0.0 (0.0–0.1) [0.1–2.1] | 0.5 (0.3–0.7) [0.1–2.1] |
| Platelets (×10$^9$/L) | 218 (114–273) [186–545] | 234 (191–274) [186–545] |
| Albumin (g/L) | 25 (22–31) [32–41] | 3.9 (3.7–4.0) [3.2–4.1] |
| ALT (U/L) | 50 (27–104) [17–95] | 45 (36–59) [17–95] |
| Total bilirubin (μmol/L) | 1.7 (1.7–6.8) [0.0–3.4] | 0.0 (0.0–0.0) [0.0–0.2] |
| BUN (mmol/L) | 4.3 (3.2–7.5) [3.2–9.3] | 6.1 (5.4–7.1) [3.2–9.3] |
| Creatinine (μmol/L) | 106 ± 111 [53–124] | 97 (80–97) [53–124] |

Data are presented as mean ± standard deviation for normally distributed data and median (interquartile range) for non-normally distributed data. Reference intervals are presented in square parentheses [].

Table 2. Medications prescribed to dogs with sepsis prior to study enrolment.

| Medication class | n |
| --- | --- |
| Antimicrobial drugs (AMD)<br>• Beta-lactams (n = 10)<br>• Fluoroquinolones (n = 6)<br>• Nitroimidazoles (n = 5)<br>• Tetracyclines (n = 1)<br>• Unknown AMD (n = 1) | 23 |
| Antiemetics / Gastroprotectants | 7 |
| Non-steroidal anti-inflammatory drugs | 6 |
| Other analgesics | 2 |
| Glucocorticoids | 1 |
| Anxiolytics | 1 |

20 healthy control dogs consisted of 12 mixed breed dogs and 8 purebred dogs including 2 mastiffs and 1 dog of each of the following breeds: akita, Bernese Mountain dog, golden retriever, Labrador retriever, redbone coonhound, rottweiler. The healthy dogs consisted of 13 spayed females, 6 castrated males and 1 intact male dog. In the healthy dog population, 19/20 dogs were eating commercial cooked diets, with 3 dogs eating more than one type of commercial food, and 1 dog was fed a home-prepared raw food diet. Two healthy dogs were fed nutritional supplements intended to maintain joint health including glucosamine and chondroitin, green lipped mussel extract, methyl sulfonyl methane, hyaluronic acid, ascorbic acid, and DL-phenylalanine. None of the dogs sampled for this study were intentionally fasted prior to sample collection. Some dogs with sepsis were inappetent for variable amounts of time prior to sampling, however.

## Untargeted metabolomics C18

The OPLS-DA score plot of the metabolomes of dogs with sepsis and healthy controls from C18 datasets demonstrated tight clustering and clear separation of the data from both groups and QC samples (Fig 2A). The QC samples were also highly repeatable and well demarcated from both groups of dogs. After automatic and manual curation to identify and remove likely drug metabolites and duplicate identifications, 198 compounds were identified by C18, positive mode analysis, and 349 compounds identified by C18, negative mode analysis. Combining these lists and removing duplicates produced a list of 536 unique compounds identified by C18 analysis (S2 Data). Volcano plots (Fig 2B) were constructed to analyze these compounds for fold change and statistical significance having fold change >2.0 and FDR P-values <0.05 and identified groups of compounds that were both significantly increased (n = 281) and decreased in sepsis (n = 127) compared to healthy controls. A substantial number (n = 147) of compounds were not significantly different between the two groups. Heatmaps were constructed to enable visualization of data for the top 25 highly significant compounds that had large changes in abundance in dogs with sepsis at the lowest P-values <0.001 (Fig 2C).

Small molecule pathway database (SMPDB) metabolic pathway analysis (performed in MetaboAnalyst 5.0) for filtered annotated compounds in both C18 modes were ranked by P-value. This identified 4 major metabolic pathways that were 20- to 30-fold enriched in the metabolome with a corresponding P-value <0.05 ($-\log_{10}P$ >1.3). These pathways were pyruvaldehyde degradation, ketone body metabolism, the glucose-alanine cycle and Vitamin-K metabolism (Fig 3A). Pathway analysis using the Kyoto Encyclopedia of Genes and Genomes (KEGG) database yielded identification of 5 pathways that were 2- to 6-fold enriched in the metabolome: phenylalanine, tyrosine and tryptophan biosynthesis; aminoacyl-tRNA

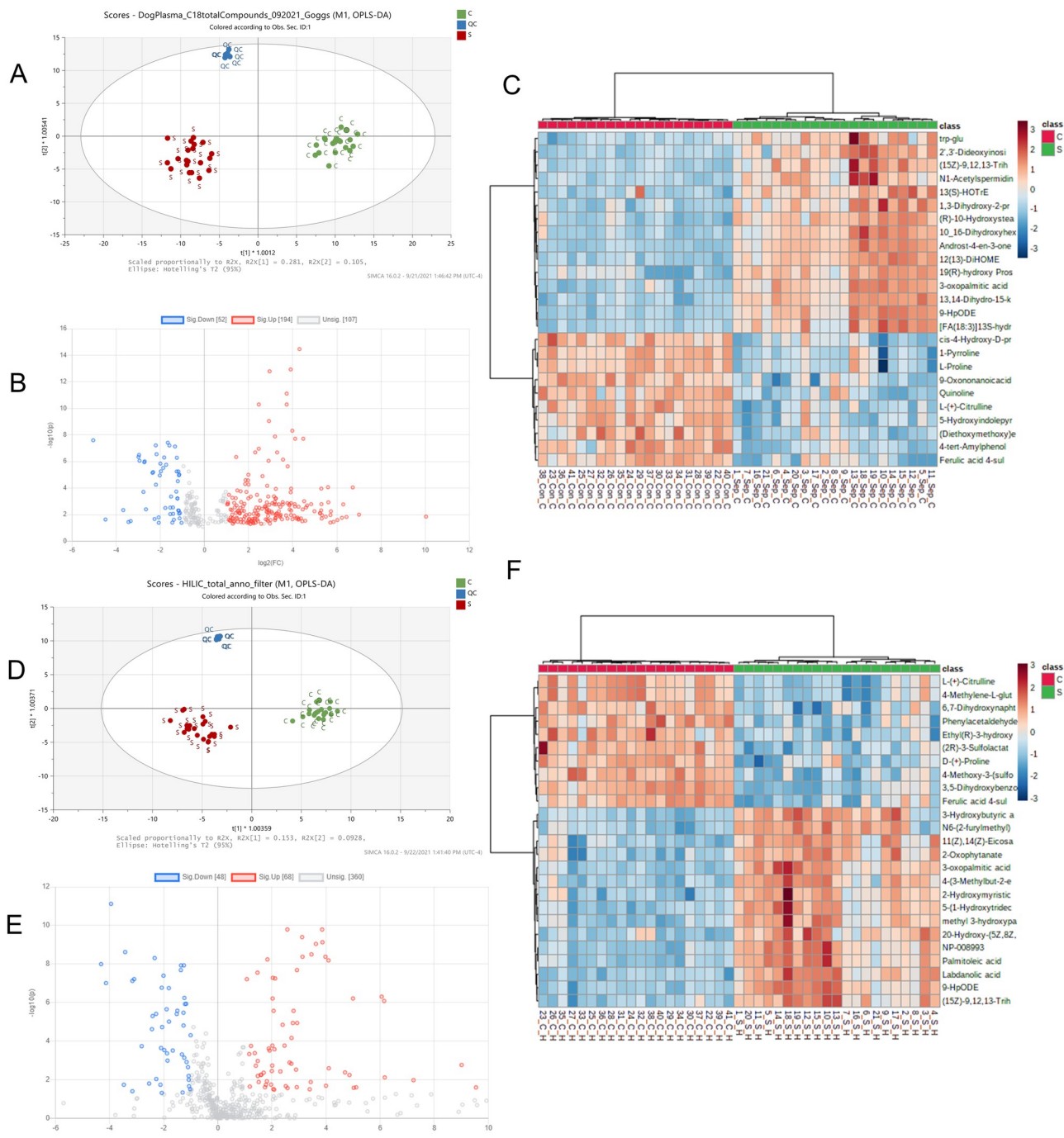

**Fig 2. Untargeted metabolomic analyses of plasma samples from dogs with sepsis compared to healthy controls.** (A) OPLS-DA plot for C18 metabolites. $R^2X = 0.608$, $R^2Y = 0.934$, $Q^2 = 0.892$. (B) Volcano plots for C18 metabolites. P-value <0.05, FC = S/C. (C) Heatmap: Top 25 significant metabolites in C18 (total 536 unique metabolites in C18 positive and negative modes). (D) OPLS-DA plot for HILIC metabolites. $R^2X = 0.472$, $R^2Y = 0.954$, $Q^2 = 0.879$. (E) Volcano plots for HILIC metabolites. P<0.05, FC = S/C. (F) Top 25 significant metabolites in HILIC (total 386 unique metabolites in HILIC positive and negative modes).

biosynthesis, valine, leucine and isoleucine biosynthesis; arginine and proline metabolism; phenylalanine metabolism. Of these, only phenylalanine, tyrosine and tryptophan biosynthesis were significantly enriched (P<0.05) (Fig 3B).

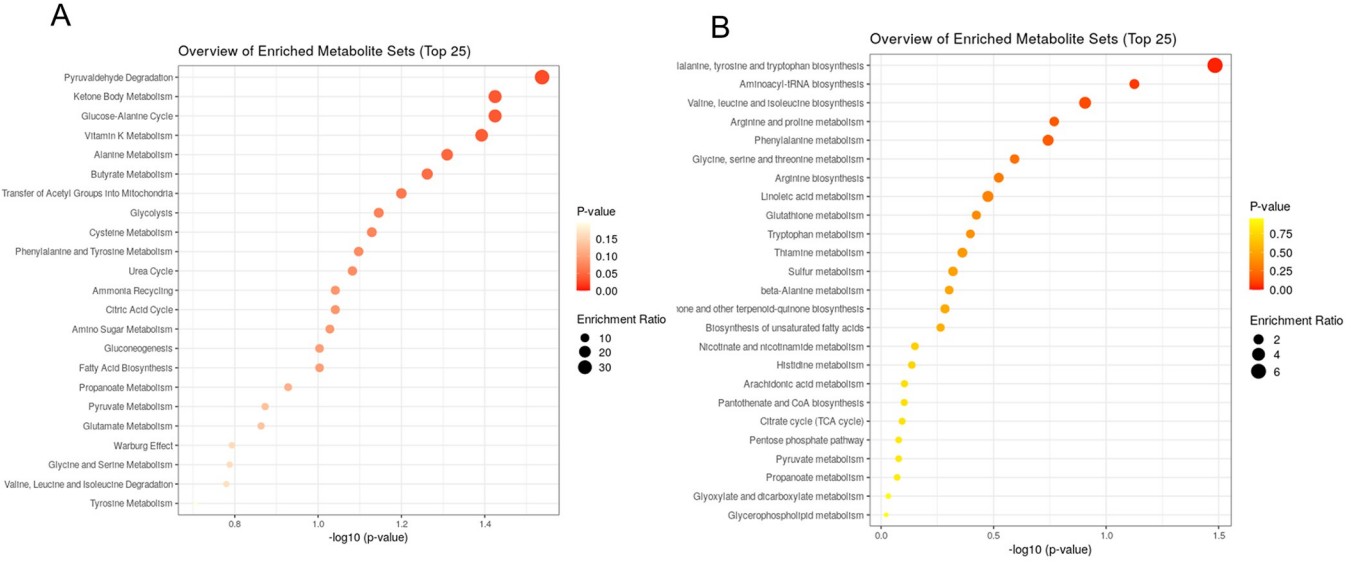

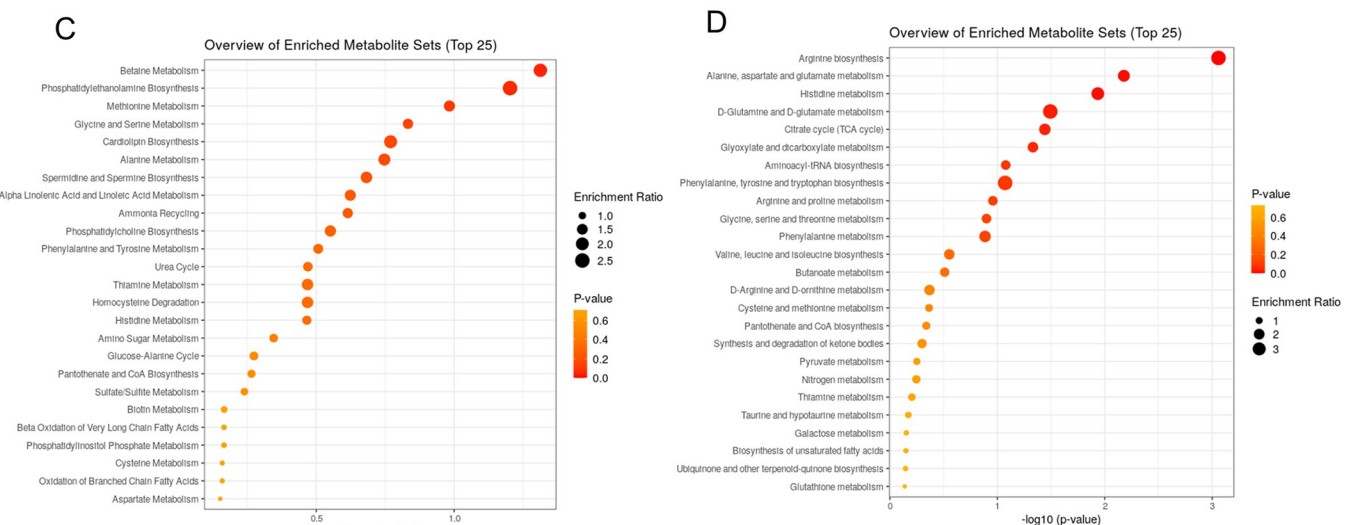

**Fig 3. Metabolomics pathway analyses.** (A) Small molecule pathway database (SMPDB) metabolic pathway analysis for filtered annotated compounds in both C18 modes ranked by P-value. (B) Kyoto Encyclopedia of Genes and Genomes (KEGG) metabolic pathway analysis for filtered annotated compounds in both C18 modes ranked by P-value. (C) Small molecule pathway database (SMPDB) metabolic pathway analysis for filtered annotated compounds in both HILIC modes ranked by P-value. (D) Kyoto Encyclopedia of Genes and Genomes (KEGG) metabolic pathway analysis for filtered annotated compounds in both HILIC modes ranked by P-value.

## Untargeted metabolomics HILIC

As for C18 analysis, OPLS-DA score plot of the metabolomes of dogs with sepsis and healthy controls from HILIC datasets also demonstrated tight clustering and clear separation of dogs with sepsis from healthy controls, and with distinct representation of QC data (Fig 2D). After automatic and manual curation to identify likely drug metabolites and duplicate identifications, 164 compounds were identified by HILIC in positive ion mode, and 226 compounds identified in negative mode. Combining these lists and removing duplicate annotations

produced a list of 476 unique compounds identified by HILIC analysis (S2 Data). Volcano plots (Fig 2E) showed 48 compounds that were significantly increased having fold change >2.0 and FDR P-values <0.05, 68 compounds that were decreased in sepsis and 360 compounds that were not significantly different between the two groups. Heatmaps were constructed to enable visualization of data for the 25 compounds with the lowest P-values <0.001 (Fig 2F). In total across both C18 and HILIC analyses, 803 unique compounds were identified.

Within the HILIC dataset, pathway analysis using the SMPDB approach identified two pathways that were 2- to 2.5-fold enriched within the metabolome with associated P-values <0.05: betaine metabolism and phosphatidylethanolamine metabolism (Fig 3C). Evaluation of the HILIC dataset using KEGG pathway analysis identified 5 pathways that were significantly (P<0.05) enriched by 2- to 3-fold within the metabolome: arginine biosynthesis; alanine and aspartate metabolism; histidine metabolism; glutamine/glutamate metabolism and the citric acid (tricarboxylic acid) cycle (Fig 3D).

### Comparisons with human sepsis

To contextualize the dog sepsis metabolome data using previously reported data on the metabolome in human sepsis metabolome data we compared our C18 and HILIC data with publicly available data from a study of the metabolomes of 197 critically ill humans with early sepsis [29]. Specifically, we compared combined curated C18 and HILIC data (filtered based on MS2 spectra confirmation and curated to remove known exposome compounds, plasticizers, and drug metabolites) with those from Rogers et al. [29]. We identified 107 metabolites (predominantly amino acids, fatty acids, lipids classes including sphingomyelins and phosphocholines, and TCA metabolites like lactate, citrate, oxoglutarate) that were common between dogs and humans (Fig 4, S3 Data).

### Potential biomarkers for sepsis diagnosis

Two approaches were used to identify putative biomarkers for sepsis using combined data from both C18 and HILIC analyses. Tabulated data detailing fold-change and corresponding statistical significance were used to identify compounds with the greatest product of $\log_2$[fold-change] and -log[P-value] for compounds with increased abundance in sepsis (Table 3) and of -$\log_2$[fold-change] and -log[P-value] for compounds with decreased abundance in sepsis (Table 4). These compounds correspond to the points in the upper-right and upper-left parts of the volcano plots. In addition, receiver operating characteristic curves with corresponding box-whisker discrimination plots were generated for highly statistically significant metabolites. Multiple compounds showed high discriminant abilities (Fig 5) with AUC values of ROC plots above 0.8. Three compounds with increased abundance in sepsis had AUC values of 1.0: 13,14-Dihydro-15-keto Prostaglandin $A_2$ with a fold-change of 7.73, P = $1.38 \times 10^{-15}$; 12(13)-DiHOME (12,13-dihydroxy-9Z-octadecenoic acid) with a fold-change of 15.0, P = $6.62 \times 10^{-16}$; and 9-HpODE (9-Hydroperoxyoctadecadienoic acid) with a fold-change of 13.2, P = $8.77 \times 10^{-14}$.

### Potential prognostic biomarkers

Metabolomic profiles of the 3 non-survivors were compared with those of the 18 dogs that were discharged alive to identify potential biomarkers associated with outcome (S2 Data). Principal component analysis of data from both C18 and HILIC analyses clearly demarcated survivors from non-survivors (Figs 6 and 7). Similarly, heatmap analyses and volcano plots suggested that numerous metabolites were increased in abundance in non-survivors compared to survivors (Figs 6 and 7), with a smaller number of compounds increased in survivors (i.e.,

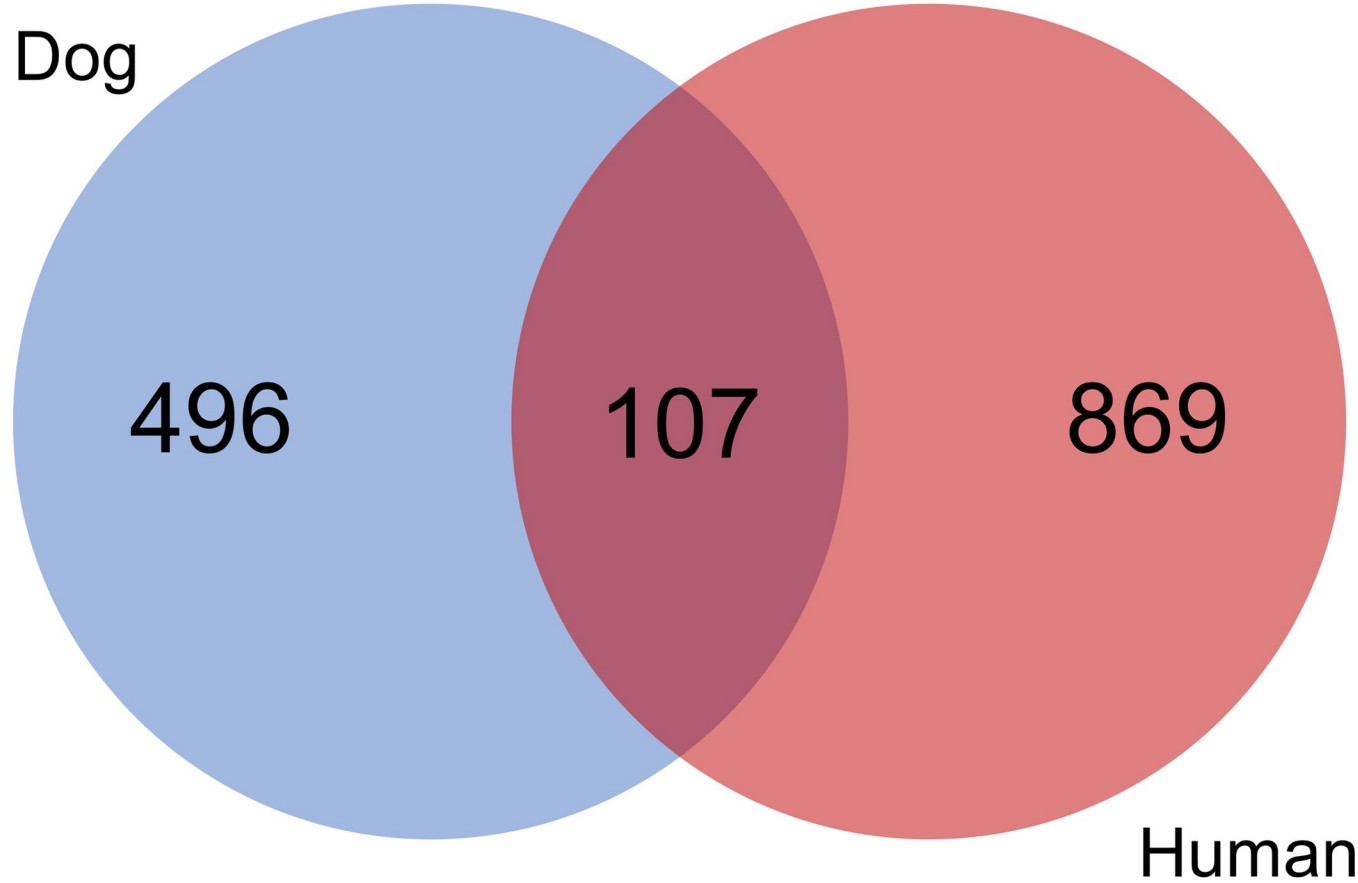

**Fig 4. Venn diagram comparing metabolites in human and canine sepsis.** Comparisons between metabolites identified in a study of humans with early sepsis and curated combined data from C18 and HILIC analyses in dogs identified 107 compounds in common, with 496 compounds unique to dogs, and 869 compounds unique to humans.

decreased in dogs that were euthanized). Analysis of metabolites up-regulated in nonsurvivors identified several compounds derived from intestinal microbiota tryptophan metabolism including 3-(2-hydroxyethyl) indole, indoxyl sulfate and xanthurenic acid and various compounds derived from bacterial metabolism including p-Cresol glucuronide and 2-aminoacetophenone. Compounds down-regulated in nonsurvivors included O-feruloylquinate, thiamine,

**Table 3. Top 10 compounds upregulated in dogs with sepsis.**

| Compound | FC | Log$_2$FC | P-value | -Log P | Log$_2$FC × -Log P |
|---|---|---|---|---|---|
| Androst-4-en-3-one | 19.764 | 4.3058 | $9.75 \times 10^{-18}$ | 17.01 | 73.23 |
| 12(13)-DiHOME | 15.000 | 3.907 | $6.62 \times 10^{-16}$ | 15.18 | 59.30 |
| 9-HpODE | 13.224 | 3.725 | $8.77 \times 10^{-14}$ | 13.06 | 48.64 |
| 3-oxopalmitic acid | 13.232 | 3.726 | $8.03 \times 10^{-13}$ | 12.10 | 45.07 |
| 13,14-Dihydro-15-keto Prostaglandin A$_2$ | 7.732 | 2.951 | $1.38 \times 10^{-15}$ | 14.86 | 43.85 |
| 2',3'-Dideoxyinosine | 22.241 | 4.475 | $6.05 \times 10^{-10}$ | 9.22 | 41.25 |
| 19(R)-hydroxy Prostaglandin A$_2$ | 15.788 | 3.981 | $1.10 \times 10^{-10}$ | 9.96 | 39.64 |
| (15Z)-9,12,13-Trihydroxy-15-octadecenoic acid | 17.249 | 4.108 | $5.94 \times 10^{-10}$ | 9.23 | 37.91 |
| 10,16-Dihydroxy hexadecenoic acid | 7.579 | 2.922 | $1.79 \times 10^{-11}$ | 10.75 | 31.40 |
| 13S-hydroperoxy-9Z,11E,14Z-octadecatrienoic acid | 9.157 | 3.195 | $3.34 \times 10^{-10}$ | 9.48 | 30.27 |

**Table 4. Top 10 compounds downregulated in dogs with sepsis.**

| Compound | FC | 1/FC | Log₂FC | P-value | -Log₁₀ P | -Log₂FC × -Log₁₀ P |
|---|---|---|---|---|---|---|
| 4-tert-Amylphenol | 0.030 | 32.918 | 5.041 | $8.63 \times 10^{-10}$ | 9.06 | 45.69 |
| Quinoline | 0.128 | 7.810 | 2.965 | $1.91 \times 10^{-8}$ | 7.72 | 22.89 |
| Ferulic acid 4-sulfate | 0.126 | 7.948 | 2.991 | $3.00 \times 10^{-8}$ | 7.52 | 22.50 |
| 2-Aminoindan-2-carboxylic acid | 0.130 | 7.687 | 2.942 | $6.81 \times 10^{-8}$ | 7.17 | 21.09 |
| 8-Hydroxyquinoline | 0.153 | 6.545 | 2.710 | $8.05 \times 10^{-8}$ | 7.09 | 19.23 |
| 2,4,6-Trihydroxy benzophenone | 0.151 | 6.626 | 2.728 | $9.82 \times 10^{-8}$ | 7.01 | 19.12 |
| cis-4-Hydroxy-D-proline | 0.217 | 4.619 | 2.208 | $2.43 \times 10^{-9}$ | 8.61 | 19.02 |
| 9-Oxononanoicacid | 0.222 | 4.505 | 2.171 | $1.61 \times 10^{-8}$ | 7.79 | 16.92 |
| L-(+)-Citrulline | 0.316 | 3.161 | 1.661 | $1.41 \times 10^{-9}$ | 8.85 | 14.70 |
| 4-Vinylpyridine | 0.195 | 5.121 | 2.356 | $8.38 \times 10^{-7}$ | 6.08 | 14.32 |

5-phenyl nicotinic acid, cytosine, 1,5-Anhydro-D-glucitol and arachidonic acid. To further assess the utility of these biomarkers, non-parametric correlations between the clinical illness severity score (APPLE$_{fast}$) and the abundances of the top 50 biomarkers for both C18 and HILIC analyses were calculated (S1 Fig). Multiple prognostic biomarkers were significantly associated with illness severity, suggesting that prognostic metabolomic biomarkers identified likely reflect the severity of sepsis in dogs.

## Untargeted lipidomics

In positive mode, C30 LC-MS/MS analyses identified 323 lipids including isomers. The predominant lipid class was triglycerides (TG), with phosphatidylcholine (PC) isomers and sphingomyelins (SM) the two other major classes (Fig 8A). OPLS-DA score plots of the lipidomes of

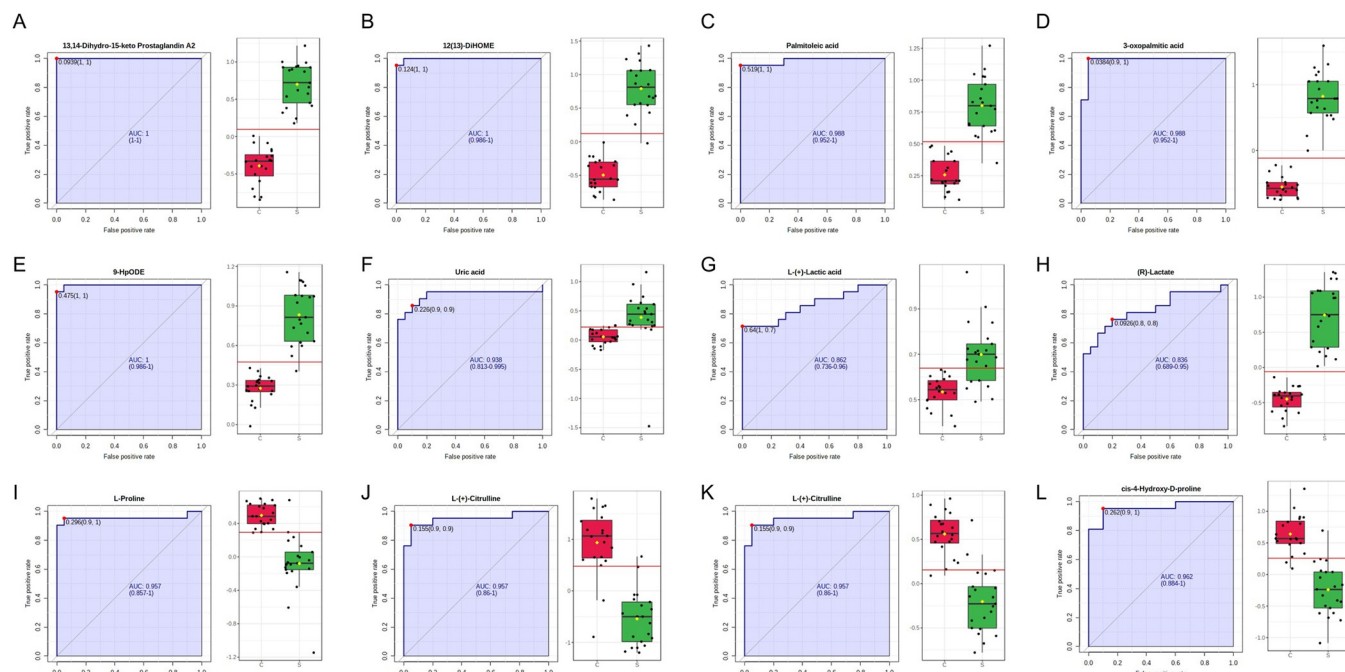

**Fig 5. ROC plots of putative biomarkers for sepsis.** (A-H) ROC plots of putative positive biomarkers for sepsis. (I-L) ROC plots of putative negative biomarkers for sepsis. Corresponding box-whisker plots for optimal cutoffs are displayed in tandem with each ROC curve.

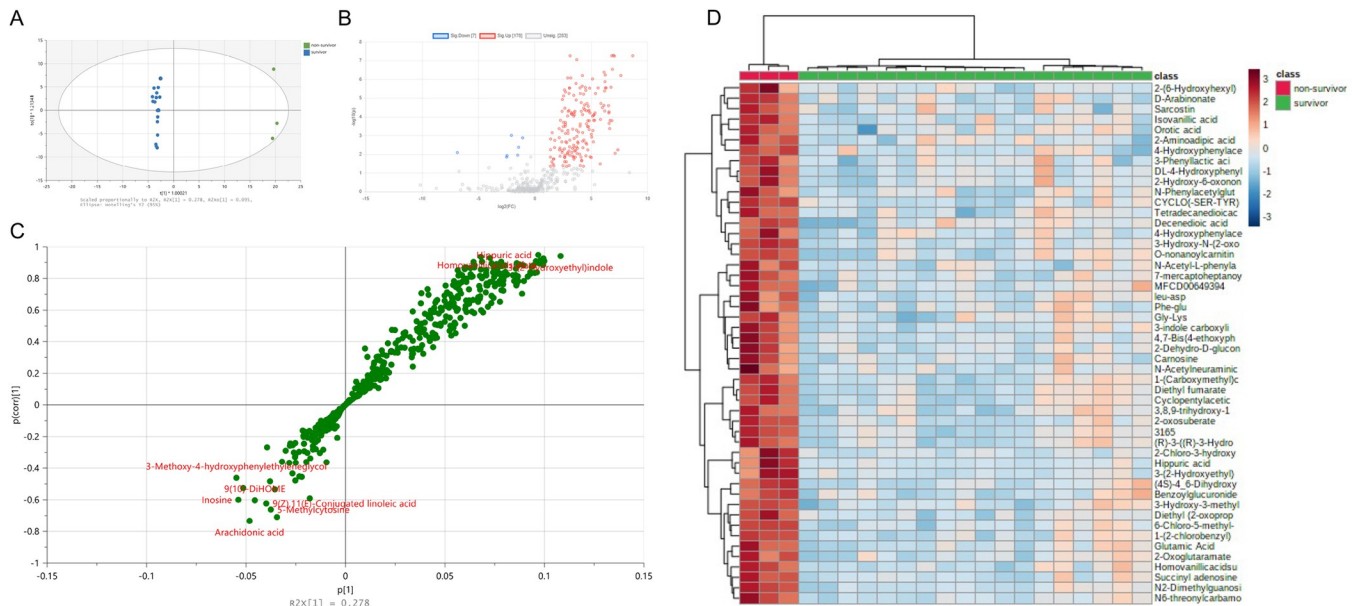

**Fig 6. Identification of potential prognostic biomarkers for sepsis in dogs using untargeted metabolomics (C18 analyses).** (A) OPLS-DA plot for C18 metabolites. $R^2X = 0.442$, $R^2Y = 0.997$, $Q^2 = 0.859$. (B) Volcano plots for C18 metabolites. Nonsurvivor / survivor. P-value $<0.05$ FDR, $\log_2$(fold change)$>2.0$. (C) S-plot of metabolites identified by C18 analysis and their modeled class designation. (D) Heatmap: Top 50 significant metabolites in C18 after log transformation and pareto scaling.

dogs with sepsis and healthy controls from the C30 lipidome dataset acquired in positive mode demonstrated adequate clustering and clear separation of the data from both groups (Fig 8B). Multivariate and significant features detection analyses identified various SM forms and

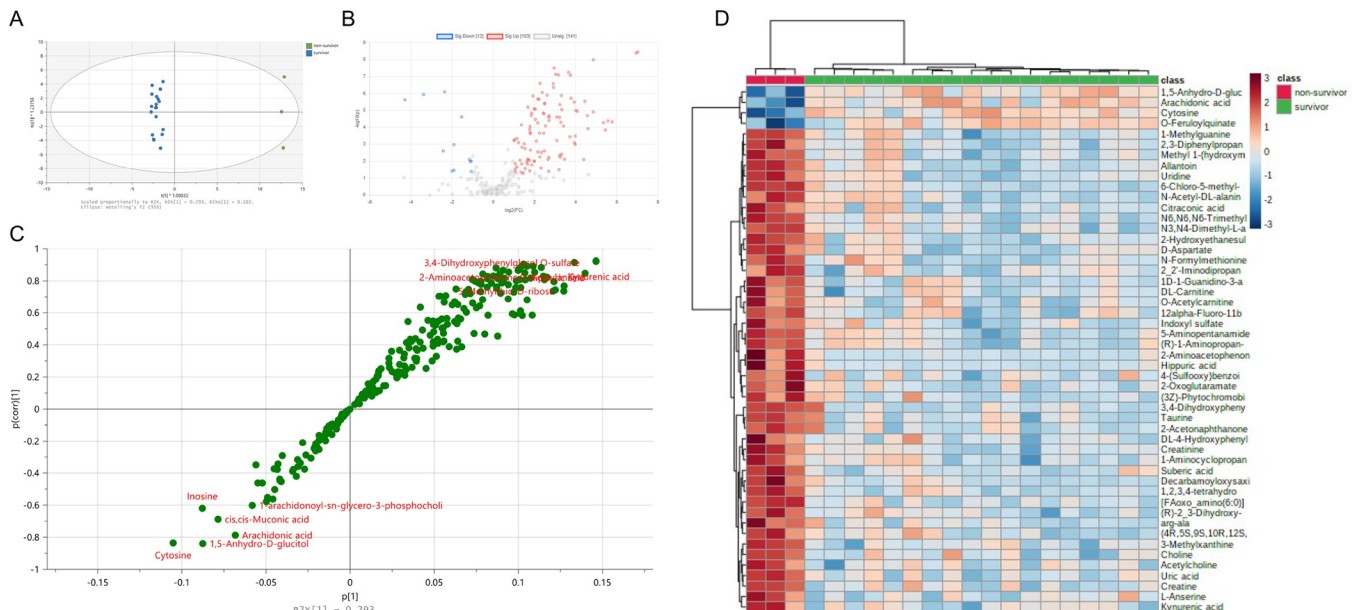

**Fig 7. Identification of potential prognostic biomarkers for sepsis in dogs using untargeted metabolomics (HILIC analyses).** (A) OPLS-DA plot for HILIC metabolites. $R^2X = 0.453$, $R^2Y = 0.993$, $Q^2 = 0.893$. (B) Volcano plots for HILIC metabolites. Nonsurvivor / survivor. P-value $<0.05$ FDR, $\log_2$(fold change)$> 2.0$. (C) S-plot of metabolites identified by HILIC analysis and their modeled class designation. (D) Heatmap: Top 50 significant metabolites in HILIC after log transformation and pareto scaling.

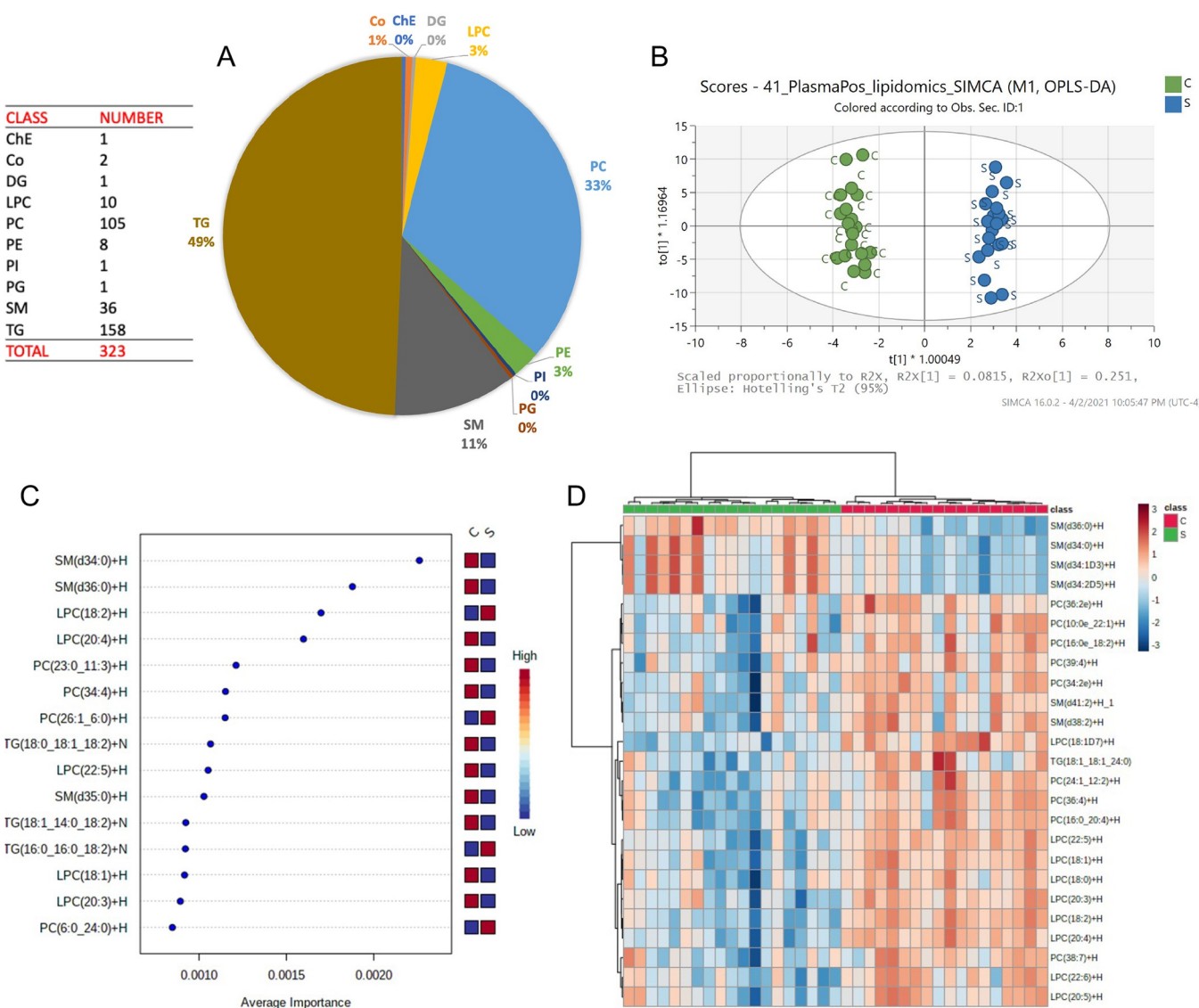

**Fig 8. Lipids identified in positive mode 323 lipids (inc. isomers after filtering data).** Highest amount corresponded to TG. C30 positive mode: OPLS-DA plot and Heatmap. Multivariate and significant features detection analysis. Compounds detected in C30 positive mode ranked by average importance.

lysophosphatidylcholines (LPCs) that were discriminating for sepsis or for controls (Fig 8C). Heatmaps were constructed to enable visualization of data for the 25 lipids with the lowest P-values (Fig 8D). Tabulated data detailing relative abundance in dogs with sepsis and healthy controls were used to identify compounds up- and down-regulated in sepsis (Table 5). The top 10 lipids most increased in abundance in dogs with sepsis were a mixture of lipid classes, predominantly TGs composed of saturated or monounsaturated fatty acids and PCs. The 10 lipids most decreased in abundance in dogs with sepsis were all TG consisting primarily of polyunsaturated fatty acids (Table 5).

In negative mode, C30 LC-MS/MS analyses identified 183 lipids including isomers. The predominant lipid class was phosphatidylcholine (PC) isomers, with phosphatidylethanolamines (PE) and sphingomyelins (SMs) constituting the two other major classes (Fig 9A). OPLS-DA score plots of the lipidomes of dogs with sepsis and healthy controls demonstrated

**Table 5. Top 10 lipids upregulated and downregulated in dogs with sepsis C30 positive ion mode.**

| Lipid | Fatty Acids | Ion Formula | Relative abundance S/C |
|---|---|---|---|
| TG(50:2)+NH4 | (16:0_16:0_18:2) | C53 H102 O6 N1 | 5.880 |
| TG(50:3)+NH4 | (18:1_14:0_18:2) | C53 H100 O6 N1 | 5.434 |
| SM(d36:0)+H | (d36:0) | C41 H86 O6 N2 P1 | 5.028 |
| PE(34:1)+H | (34:1) | C39 H77 O8 N1 P1 | 4.660 |
| PC(28:0)+H | (28:0) | C36 H73 O8 N1 P1 | 4.257 |
| PC(38:3)+H | (38:3) | C46 H87 O8 N1 P1 | 3.905 |
| TG(46:1)+NH4 | (16:0_12:0_18:1) | C49 H96 O6 N1 | 3.793 |
| PC(30:0)+H | (6:0_24:0) | C38 H77 O8 N1 P1 | 2.868 |
| TG(49:2)+NH4 | (18:1_13:0_18:1) | C52 H100 O6 N1 | 2.856 |
| TG(48:0)+NH4 | (16:0_16:0_16:0) | C51 H102 O6 N1 | 2.709 |
| TG(50:5)+NH4 | (18:2_14:1_18:2) | C53 H96 O6 N1 | 0.278 |
| TG(62:3)+NH4 | (26:0_18:1_18:2) | C65 H124 O6 N1 | 0.206 |
| TG(60:2)+NH4 | (18:1_18:1_24:0) | C63 H122 O6 N1 | 0.170 |
| TG(58:2)+NH4 | (16:0_18:2_24:0) | C61 H118 O6 N1 | 0.169 |
| TG(60:3)+NH4 | (18:1_18:2_24:0) | C63 H120 O6 N1 | 0.143 |
| TG(54:8)+NH4 | (18:3_18:2_18:3) | C57 H98 O6 N1 | 0.129 |
| TG(58:3)+NH4 | (18:1_18:2_22:0) | C61 H116 O6 N1 | 0.125 |
| TG(54:7)+NH4 | (18:3_18:2_18:2) | C57 H100 O6 N1 | 0.123 |
| TG(60:4)+NH4 | (24:0_18:2_18:2) | C63 H118 O6 N1 | 0.100 |

adequate clustering and clear separation of the data from both groups (Fig 9B). Multivariate and significant features detection analyses identified two SM isomers were particularly discriminating for sepsis or for controls (Fig 9C). Heatmaps were constructed to enable visualization of data for the 25 compounds with the lowest P-values (Fig 9D). Through analysis of tabulated data from negative ion mode analysis, the 10 lipids most increased in dogs with sepsis were predominantly PC and PE compounds typically containing saturated or monounsaturated fatty acids. The 10 lipids as most decreased in dogs with sepsis were predominantly PC and LPC compounds consisting primarily of polyunsaturated fatty acids (Table 6).

## Discussion

The heterogeneous nature of sepsis in dogs presents a diagnostic and prognostic challenge for both clinicians and researchers. Novel treatment strategies are needed to improve outcomes. Metabolomics is an emerging tool for studying the pathophysiology of sepsis. Metabolomics might also yield therapeutic opportunities and identify discriminating biomarkers for determining the diagnosis and prognosis of sepsis in humans [30–32]. In rodent models of sepsis, concentrations of a panel of lipid metabolites were shown to be highly predictive of survival. These metabolites included: linolenic acid, linoleic acid, oleic acid, stearic acid, docosahexaenoic acid and docosapentaenoic acid [30]. The study also revealed that a panel of metabolites, predominantly involved in energy metabolism, were prognostic in 87% of animals. These metabolites included lactate, alanine, acetate, acetoacetate, hydroxybutyrate, and formate [31]. Increased serum concentrations of alanine, creatine, phosphoethanolamine, and acetoacetate and decreased concentrations of formate are highly sensitive and specific for the diagnosis of sepsis in rodents [32]. Metabolomic and lipidomic profiling of humans with sepsis has identified potential diagnostic biomarkers and distinct metabolomic patterns with the potential to guide clinical decision-making [13, 33–35]. For instance, in a case-control study, the metabolomic profiles of humans with septic shock had significant increases in phenylalanine, myo-

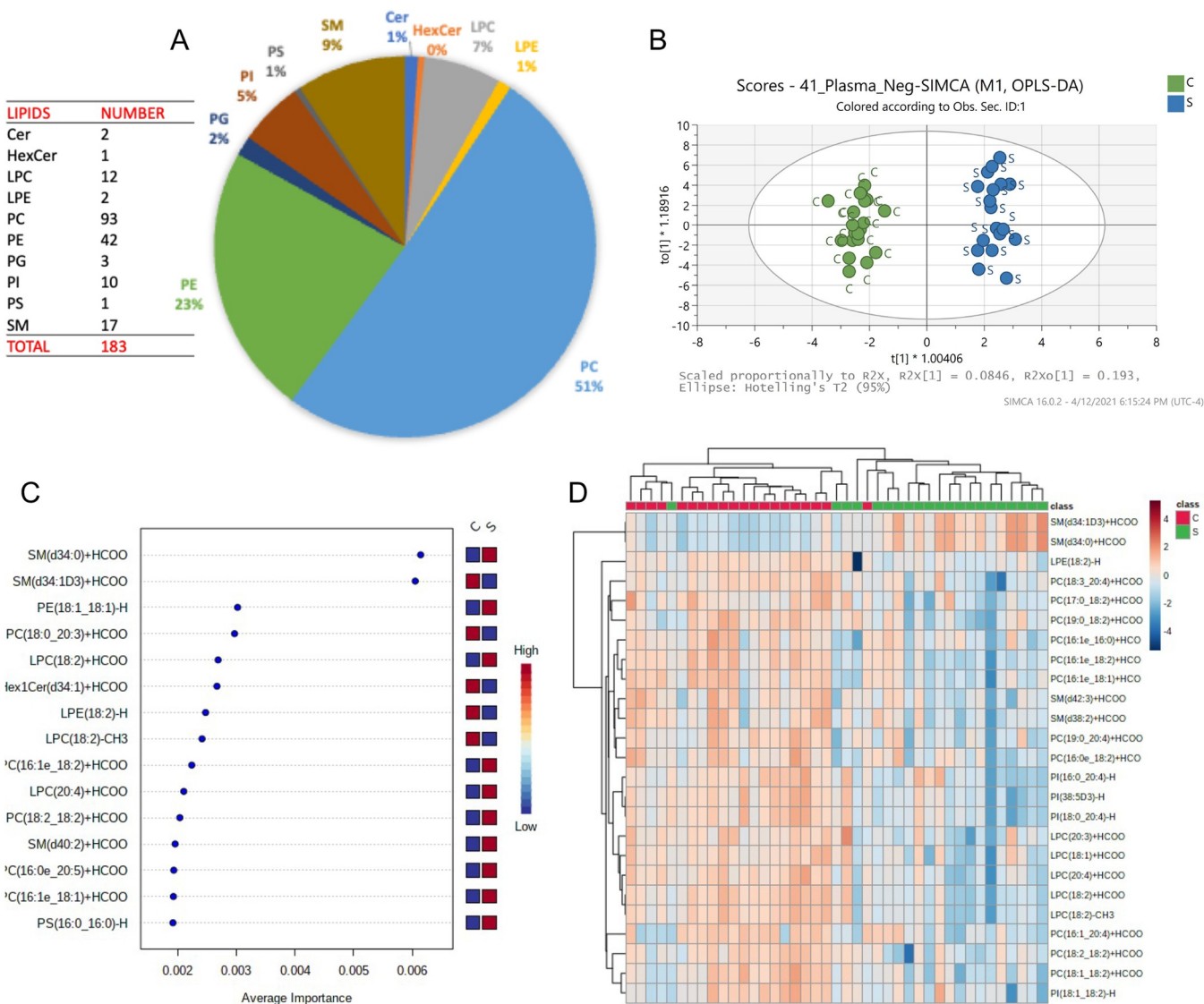

**Fig 9. Lipids identified in negative mode 183 lipids (inc. isomers after filtering data).** Highest amount corresponded to PC. C30 negative mode: OPLS-DA plot and Heatmap. Multivariate and significant features detection analysis. Compounds detected in C30 negative mode ranked by average importance.

inositol, isobutyrate and hydroxybutyrate and reductions in the amino acids arginine, valine, and threonine, as compared to human ICU patients with non-septic SIRS [33]. Furthermore, increased concentrations of 2-hydroxyisovalerate and fructose, and decreased concentrations of dimethylamine were associated with nonsurvival. In a pediatric sepsis cohort, metabolomic profiles outperformed procalcitonin measurements and the Pediatric Risk of Mortality III-Acute Physiology Score for discrimination of septic shock from non-septic SIRS and for survival prediction [13].

Comparable metabolic profiling in dogs with sepsis has so far been limited. To address this knowledge-gap we employed metabolomics to identify diagnostic biomarkers and altered metabolic pathways that could serve as future therapeutic targets. We identified 803 unique metabolites in plasma samples from dogs with sepsis. Within those 803 compounds, 329 metabolites were identified that were significantly increased in dogs with sepsis compared to healthy

**Table 6. Top 10 lipids upregulated and downregulated in dogs with sepsis C30 negative ion mode.**

| Lipid | Fatty Acids | Ion Formula | Relative abundance S/C |
|---|---|---|---|
| PC(28:0)+HCOO | (14:0_14:0) | C37 H73 O10 N1 P1 | 18.879 |
| PE(34:3)-H | (16:0_18:3) | C39 H71 O8 N1 P1 | 12.836 |
| PE(34:1)-H | (16:0_18:1) | C39 H75 O8 N1 P1 | 6.070 |
| PE(35:1)-H | (17:0_18:1) | C40 H77 O8 N1 P1 | 5.959 |
| PC(34:3)+HCOO | (16:1_18:2) | C43 H79 O10 N1 P1 | 5.070 |
| PE(38:3)-H | (18:0_20:3) | C43 H79 O8 N1 P1 | 3.380 |
| PC(34:2)+HCOO | (16:1_18:1) | C43 H81 O10 N1 P1 | 3.011 |
| PC(36:3)+HCOO | (16:0_20:3) | C45 H83 O10 N1 P1 | 3.011 |
| PE(38:4)-H | (16:0_22:4) | C43 H77 O8 N1 P1 | 2.908 |
| PG(36:1)-H | (18:0_18:1) | C42 H80 O10 N0 P1 | 2.803 |
| LPE(18:2)-H | (18:2) | C23 H43 O7 N1 P1 | 0.445 |
| LPC(20:4)+HCOO | (20:4) | C29 H51 O9 N1 P1 | 0.442 |
| PC(36:6e)+HCOO | (16:1e_20:5) | C45 H79 O9 N1 P1 | 0.414 |
| PC(40:5)+HCOO | (18:1_22:4) | C49 H87 O10 N1 P1 | 0.368 |
| LPC(18:2)+HCOO | (18:2) | C27 H51 O9 N1 P1 | 0.364 |
| LPC(18:2)-CH3 | (18:2) | C25 H47 O7 N1 P1 | 0.358 |
| PC(40:6)+HCOO | (18:1_22:5) | C49 H85 O10 N1 P1 | 0.345 |
| PC(36:4)+HCOO | (18:2_18:2) | C45 H81 O10 N1 P1 | 0.330 |
| PC(38:6)+HCOO | (18:1_20:5) | C47 H81 O10 N1 P1 | 0.315 |
| PC(36:5)+HCOO | (16:1_20:4) | C45 H79 O10 N1 P1 | 0.198 |

controls, whereas 195 were decreased in dogs with sepsis. Pathway analysis identified multiple enriched metabolic pathways including pyruvaldehyde degradation; ketone body metabolism; the glucose-alanine cycle; vitamin-K metabolism; arginine and betaine metabolism; the biosynthesis of various amino acid classes including the aromatic amino acids (phe, trp, tyr); branched chain amino acids (BCAA) (ile, leu, val); and metabolism of glutamine/glutamate and the glycerophospholipid phosphatidylethanolamine.

Many of the metabolites and metabolic pathways up- and down-regulated in dogs in our study have been identified as molecules or pathways of interest in studies of humans with sepsis or septic shock [29]. Overall, the metabolomic profiles of dogs with sepsis suggest a cellular energy crisis with consequent mobilization of resources to address a negative energy balance and maintain blood glucose concentrations through upregulation of the tricarboxylic acid (TCA) cycle and ongoing protein and fat catabolism [36]. Enrichment of TCA cycle intermediates such as pyruvate, proline, glutamine, glutamate, phenylalanine, and alanine likely represent host attempts to enhance energy production and might be associated with cytopathic hypoxia that is associated with poor outcomes in sepsis [37]. The profiles also suggest the potential for hepatocellular injury and alterations in the handling of Vitamin K. Several potential bacterial-derived metabolites, such as R-lactate were represented in the dog metabolomes and certain pathways including glutamine/glutamate metabolism were upregulated, suggesting dysregulation of host-microbial interactions in dogs with sepsis.

Metabolites associated with ketone body metabolism were significantly enriched in the sepsis samples, indicating enhanced fat breakdown and a tendency towards ketoacidosis, consistent with the hypercatabolic state described in sepsis [38–40]. Ketone bodies are synthesized from acetyl-coenzyme A as alternative energy sources when intracellular glucose concentrations are insufficient to meet metabolic demands. Synthesis and accumulation of acetyl-CoA and hence of ketone bodies is enhanced when glucagon concentrations increase, insulin

concentrations decrease, or insulin antagonism occurs. Glucagon also leads to catabolism through enhanced glycogenolysis, proteolysis, and lipolysis. Cytokine dysregulation is likely involved in the pathophysiology of these processes leading to enhanced ketone body metabolism. Concentrations of interleukin 8 (IL-8) and monocyte chemoattractant protein 1 (also known as CC motif chemokine ligand 2) are significantly higher in dogs with diabetes, whereas dogs with ketoacidosis have increased concentrations of keratinocyte chemoattractant, IL-18 and granulocyte-monocyte colony-stimulating factor [41]. Studies of dogs with sepsis have identified similar patterns of cytokine expression [42–45]. Ketone bodies also suppress the immune system [46], with increased concentrations associated with poor prognosis in human critical illness [47].

Enriched biosynthesis of the BCAA leucine, isoleucine, and valine are seen during muscle protein breakdown, consistent with the catabolic state in sepsis [48–52]. Alterations in BCAA levels have been reported in conditions resulting in systemic inflammation wherein cytokine production, sympathetic nervous system activation, and hypercortisolemia generate insulin resistance and result in muscle protein degradation and diminished uptake with a resultant loss of lean body mass [53]. This finding might support the supplementation of BCAA in dogs with sepsis to provide energy substrates for muscles, enable protein anabolism and to act as glutamine precursors to enhance immune function and maintain gut integrity [54, 55]. Upregulated histidine and phenylalanine metabolic pathways are similarly associated with muscular protein breakdown, amino acid oxidation, decreased energy supply, and organ failure seen in inflammatory states and septic shock, and have been associated with poor prognosis in critically ill humans [56–62]. Enhanced arginine pathways in humans with septic shock might be associated with nitric oxide synthesis, with increased arginine concentrations shown to impact acute phase protein synthesis [63, 64]. Increased alanine, glutamate, and phenylalanine concentrations are also seen in hemolysis associated with sepsis [65] Enriched phosphatidylethanolamine metabolism suggests dysregulated conversion of phosphatidylethanolamine to phosphatidylcholine that is associated with hepatocellular damage [66]. Enriched betaine metabolism is associated with enhanced fatty acid oxidation and export of hepatic lipids associated with fat catabolism [67].

Pathway analysis identified enrichment of glutamine/glutamate metabolism in dogs with sepsis. Glutamine is the most abundant nonessential amino acid in humans and exists in both L- and D- enantiomeric forms, whereas only L-glutamate is a metabolic intermediate in mammals. D-glutamate is not endogenously produced in humans or dogs and the substantial concentrations seen in liver and other tissues is derived from plants or from the cell walls of bacteria [68]. In our study the glutamine/glutamate pathway was upregulated, although it is not possible to be certain if this reflects the levo- or dextro- forms based solely on the mass spectra. Both L- and D-glutamine have the same mass spectra and fragment in the same way, and can only be differentiated by chiral chromatography, not reverse phase C18 chromatography as performed here. Endotoxemia in rats is associated with reduced glutamine and glutamate concentrations [69] and in humans with fatal septic shock plasma L-glutamate concentrations and plasma glutamate/glutamine ratios are low [51]. This might be due to enhanced organ glutamate and glutamine production or consumption rates [70]. In a metabolomic study of humans with sepsis [71], D-glutamine / D-glutamate metabolites were diminished in the sepsis samples which might not have been the case here. Our findings in dogs warrant replication and further investigation employing chiral chromatography to differentiate the two glutamine/glutamate enantiomers. The explanation for the up regulation of glutamine/glutamate metabolism in dogs with sepsis is uncertain but if this represents the dextroforms then it might relate to intestinal dysbiosis and an increase in absorption of bacterial metabolites or translocation of bacteria into circulation.

We also identified enriched vitamin K metabolism in dogs with sepsis. The role of vitamin K in the activation of coagulation proteins (factors II, VII, IX, X, proteins C and S) is well known. Vitamin K deficiency causes insufficient carboxylation of endothelial protein S, increasing risk for thrombosis [72], but more typically causes an increased bleeding risk that might be misdiagnosed as disseminated intravascular coagulation in critically ill humans [73]. Comparable coagulation disturbances have been well-documented in dogs with sepsis [23, 74]. Vitamin K-dependent matrix γ-carboxyglutamic protein (MGP) is involved in preventing soft tissue calcification and protects against elastic fiber degradation in the lungs and arteries [75]. During inflammation, macrophage derived matrix metalloproteinases accelerate elastic fiber degradation and increase vitamin K utilization for MGP carboxylation [76] that can result in vitamin K deficiency, MGP depletion and consequent pulmonary damage [77]. Vitamin K is also utilized in the regulation of sphingolipid metabolism [78], a process altered in dogs in our study.

Some compounds were enriched in the serum of healthy dogs relative to dogs with sepsis (i.e. diminished in abundance in dogs with sepsis). Of the 10 compounds with the lowest relative abundance in dogs with sepsis, 7 were components of the exposome [79, 80]. These compounds, including 4-tert-amylphenol, quinoline, and trihydroxybenzophenone are not naturally occurring metabolites and are found only in individuals exposed in their environments. Of the non-exposome compounds downregulated in dogs with sepsis, L-ascorbic acid 2-sulfate was the most notable. In humans, ascorbate (Vitamin C) is an essential nutrient that acts as an antioxidant that mitigates oxidative stress [81], whereas in dogs it can be synthesized endogenously [82]. In the context of human sepsis, vitamin C has been advocated as part of a metabolic resuscitation strategy combined with thiamine and glucocorticoids [83]. However, recent large clinical trials have not demonstrated a benefit of this strategy [84–86]. In dogs, vitamin C concentrations have been documented to increase in animals hospitalized in an intensive care unit [87], which is counter to the findings from dogs in our study. As such, further investigation of vitamin C concentrations in dogs with sepsis is warranted.

Multiple metabolites were identified that were highly discriminating for sepsis that could serve as potential diagnostic biomarkers. The three best performing markers were 15-keto-13,14-Dihydro-prostaglandin $A_2$; 12(13)-DiHOME (12,13-dihydroxy-9Z-octadecenoic acid); and 9-HpODE (9-Hydroxyoctadecadienoic acid). The prostaglandin metabolite 15-keto-13,14-dihydro-$PGA_2$ is derived from $PGE_2$ [88, 89], a potent inflammatory mediator generated by cyclooxygenase 2 (COX-2) conversion of arachidonic acid. Synthesis of $PGE_2$ is upregulated by COX-2 expression after stimulation by lipopolysaccharide or proinflammatory cytokines that in turn enables activation, maturation and secretion by innate immune cells such as macrophages, neutrophils, and natural killer cells. In the setting of sepsis, $PGE_2$ synthesis is induced by both Gram-negative and Gram-positive bacteria [90]. $PGE_2$ is significantly elevated in inflammation and contributes to immune suppression and increased concentrations are associated with poor prognosis in humans with sepsis [91].

Dietary plant oils containing n-6 polyunsaturated fatty acids (n-6 PUFA), such as linoleic acid, are the precursors for bioactive eicosanoids and epoxides including 9,10-epoxyoctadecenoic acid (9,10-EpOME) and 12,13-epoxyoctadecenoic acid (12,13-EpOME) [92], which are further metabolized by soluble epoxide hydrolase (sEH) to form the corresponding linoleic diols 9,10-dihydroxyoctadecenoic acid (9,10-DiHOME) and 12,13-dihydroxyoctadecenoic acid (12,13-DiHOME) [93]. These epoxides and in particular their secondary diol containing metabolites are potentially cytotoxic [94, 95]. 12,13-DiHOME is a diol containing metabolite of linoleic acid that is derived from 12,13-EpOME (isoleukotoxin) through the activity of soluble epoxide hydrolase in neutrophils. The leukotoxins are produced by activated neutrophils and high concentrations have been observed in acute respiratory distress syndrome and burns.

These leukotoxins have neutrophil chemotactic activity and modulate neutrophil respiratory burst activity [96]. In a fatal human sepsis case report, increased concentrations of various linoleic acid metabolites including 12,13-DiHOME were detected [97].

The endogenous fatty acid agonist 9-HpODE is another linoleic acid derivative than can be generated by the activity of COX, lipoxygenase, and cytochrome P450 enzymes or after lipid peroxidation by reactive oxygen species [98–100]. Linoleic acid metabolites such as 9-HpODE are thus potential markers of oxidative stress in cells and tissues in sepsis [101]. The 9-HpODE metabolite is typically rapidly reduced into dimorphecolic acid (9-HODE) which is an agonist of peroxisome proliferator-activated receptor gamma (PPARγ) [102], that can stimulate maturation of monocytes into macrophages [103] and increases plasminogen activator inhibitor type-1 expression by endothelial cells [104]. Further studies are needed to assess the specificity and sensitivity of these biomarkers in clinical practice and it would be beneficial to use a control sample of critically ill dogs with evidence of systemic inflammation but without sepsis.

In addition to identification of potential diagnostic biomarkers, we also identified various potential prognostic biomarkers. As with other markers discussed above, metabolites were both increased and decreased in abundance in samples from nonsurvivors compared to survivors. These relative changes offer potential insights into the pathophysiology of sepsis and could represent opportunities for therapeutic interventions in the future. Metabolites that were more abundant in samples from nonsurvivors included 3-(2-hydroxyethyl) indole, indoxyl sulfate and xanthurenic acid, all of which are derived from intestinal microbiota metabolism of dietary L-tryptophan, and several are known uremic toxins. Indoxyl sulfate is one such uremic toxin, that has been documented to be increased in humans with sepsis and associated kidney injury [105]. Other compounds that were increased in abundance included p-cresol glucuronide and 2-aminoacetophenone, both of which are potentially microbial-derived. Indoxyl sulfate and p-cresol compounds are produced by the colonic microbiota and can lead to progression of kidney injury [106]. Metabolites that were less abundant in samples from nonsurvivors included two B-vitamins or their derivatives (thiamine; 5-phenyl nicotinic acid). Vitamins are key effectors in many biological processes relevant to sepsis and relative vitamin deficiencies are common in humans with sepsis. Vitamin treatment has been associated with improved outcomes in some human pediatric and adult cohorts, although these benefits have not been consistently obtained [107]. The naturally occurring monosaccharide 1,5-anhydro-D-glucitol was decreased in nonsurvivors. This is noteworthy, because in humans blood concentrations of 1,5-anhydro-D-glucitol decrease during hyperglycemia, returning to normal once euglycemia is restored [108]. Hyperglycemia is potentially detrimental in critically ill humans [109], and numerous studies have investigated the use of insulin to control sepsis-associated hyperglycemia in humans [110]. The decreased abundance of arachidonic acid in nonsurvivors is also noteworthy, because this lipid mediator is the parent molecule from which numerous eicosanoid inflammatory mediators are derived [111]. Alterations in the concentrations of the bioactive mediators derived from arachidonic acid have been associated with outcome in sepsis in humans [112]. Irrespective of the similar associations with outcome documented in humans, it should be noted that these putative biomarkers are derived from comparisons of data from survivors with those of only 3 dogs that were euthanized. All 3 dogs were euthanized for disease severity, but it is uncertain if these dogs would have died if ongoing maximal intensive care had been continued. As such, it will be necessary to determine if these preliminary findings derived from a small sample size can be replicated in larger studies of dogs with sepsis, and particularly in those dogs that die of their disease despite critical care interventions.

In dogs in our study, untargeted lipidomic profiling identified 506 lipid isomers and revealed multiple sphingomyelin (SM) (SM(d34:0)+H; SM(d36:0)+H; SM(d34:0)+HCOO;

and SM(d34:1D3)+HCOO) and lysophosphatidylcholine (LPC) molecules (LPC(18:2)+H and lipophosphoserine molecules (LPS(20:4)+H) that were discriminating for dogs with sepsis. These findings are consistent with lipidomic biomarker investigations in humans with sepsis [19] and community acquired pneumonia [113]. The lipidomes of dogs with sepsis demonstrated upregulated saturated fatty acids, monounsaturated fatty acids (MUFAs), some phosphatidylcholines (PC) and phosphatidylethanolamines (PE). Downregulated compounds included polyunsaturated fatty acids (PUFAs), some phosphatidylcholines (PC), and LPC.

Downregulation of LPC is well documented in humans with sepsis [25, 114, 115], septic shock [116], and community acquired pneumonia [113, 117]. LPC is produced from the action of pro-inflammatory phospholipase $A_2$ that liberates arachidonic acid from PC. LPC can then go on to serve as a ligand for the immunoregulatory receptor G2A expressed on mature T- and B-cells. Depletion of LPC by conversion to anti-inflammatory lysophosphatidic acid might be associated with the excessive immune response seen in sepsis [19, 115] and is correlated with death [114].

In our study, some PC compounds were upregulated whereas others were downregulated; a finding consistent with lipidomic studies of humans with sepsis [118]. This finding might be due to differences in the timing of sample collection relative to the course of disease since PC levels change as sepsis progresses. In humans, increased PC concentrations might help discriminate sepsis from SIRS, potentially due to increased lipoprotein concentrations [119]. In humans with sepsis, decreased PC concentrations might be associated with death [25], an association that could be due to inadequate supply of fatty acid precursors required for PC biosynthesis secondary to dysregulated beta-oxidation [120]. In an experimental swine septic shock model, decreased PC was documented in animals with septic shock [116], hypothesized to be due to altered functionality of hepatic lipid-modifying enzymes and ongoing hepatocyte damage [121]. This association between hepatocellular injury and PC dysregulation is consistent with our metabolic profiling of dogs with sepsis that documented enriched phosphatidylethanolamine metabolism.

As in humans with sepsis, PUFAs were downregulated in dogs with sepsis. These molecules reduce T-cell activation and dampen inflammation [122], which might underpin the association between reduced PUFA concentrations and death from sepsis [25]. The reduced PUFA concentrations we observed might be due to degradation by peroxidation from reactive oxygen species or through utilization for synthesis of inflammatory mediators such as prostaglandins [123]. Upregulation of saturated fatty acids and MUFAs observed in dogs in our study is also seen in humans with sepsis and likely reflects increased lipolysis [19]. Elevated MUFA concentrations are associated with hypoalbuminemia and myocardial injury and might contribute to organ dysfunction and death in sepsis [124–127].

Our study has some limitations. We studied a heterogenous sample of dogs with sepsis, with various sources of infection and with sepsis caused by a range of different organisms. This heterogeneity is likely representative of the populations of dogs with sepsis managed by veterinarians and enhances the generalizability of our results. However, this heterogeneity could also have limited our ability to discriminate patterns within the data and to identify important but subtle differences between dogs with sepsis and controls. Similarly, the dogs with sepsis enrolled into our study were not enrolled at the same stage of their disease processes and were also affected to varying degrees by their disease process, as indicated by the wide range of disease severity scores documented in this sample. Sepsis is a syndrome characterized by the dysregulated host response to infection [1] and does not represent a single disease entity. As such, it includes a broad range of underlying causes, each of which might affect the metabolome distinctly according to the nature, duration, magnitude, and source of infection and the animal's pre-existing health status, physiologic reserves, and the impact of

intercurrent diseases. The metabolomic data we analyzed represent a single snapshot of the dogs with sepsis, yet metabolic processes are dynamic and change over time. Serial evaluations might provide a fuller picture of the metabolic alterations in septic dogs. In addition, we could not standardize the therapies, diets, or nutritional supplements that these dogs received, yet all these factors likely impacted the resulting metabolomes. Similarly, we were not able to standardize the diets of the healthy control dogs sampled and this likely also introduced variation in our data that is unrelated to sepsis. Given the heterogeneity of naturally occurring disease, it will never be possible to standardize or control for every potential contributor to a metabolic profile and indeed this might not be desirable because the data would lack real world validity. Expanding the size of the cohort evaluated could help to overcome some of these influences, however, by reducing the effect of individual variations on the average. Dietary information was available for many dogs in our study, but there was very little consistency in the type or source of the foodstuffs being consumed by these dogs, and the dogs had a range of prior nutritional statuses (history of anorexia, obesity, evidence of catabolism, etc.) that inevitably impacted the resulting metabolomic and lipidomic profiles, particularly in a comparatively small sample size. Serial measurements throughout treatment might also enhance the performance of biomarkers for monitoring response to treatment and outcome prediction. Here, we deliberately chose to compare the metabolomes of dogs with sepsis with those of healthy control animals to maximize the differences between the two groups. However, the clinical relevance of the diagnostic biomarkers we identified, and their discriminatory performance will need to be assessed in subsequent studies using other critically ill, but non-septic, cohorts of dogs.

In summary, we evaluated the metabolomic and lipidomic profiles of plasma samples from dogs with sepsis and identified numerous metabolic derangements compared to healthy control animals. Dogs with sepsis had various disruptions of metabolic pathways that were consistent with profiles of humans with sepsis and some distinct alterations that will need to be replicated. Multiple potential metabolite and lipid biomarkers were identified that could aid in the diagnosis of dogs with sepsis and could provide prognostic information or act as therapeutic targets. The substantial alterations in numerous metabolic pathways identified strongly indicates that sepsis radically alters host biochemical processes suggestive of catabolism and disrupted energy generation. Future studies should compare the metabolomic profiles of dogs with sepsis against those of other critically ill cohorts and determine if metabolomic profiling can help stratify illness severity, monitor response to treatment, or guide nutritional or other therapeutic interventions.

## Supporting information

**S1 Data. Tabulated panel of 7 internal lipid standards used as controls for lipidomics.**
(PDF)

**S2 Data. Spreadsheets containing summary metabolomics data.** The file contains a series of curated lists of metabolites identified by C18 and HILIC analyses including putative diagnostic and prognostic biomarkers.
(XLSX)

**S3 Data. Spreadsheets comparing dog and human metabolomes.** The file contains 3 sheets of curated lists of metabolites identified in dogs only, humans only and identified in both humans and dogs. Included human data derive from reference 29.
(XLSX)

**S1 Fig. Scatterplots of correlations between illness severity and select prognostic biomarkers.** Scatterplots were constructed to explore the relationships between illness severity as assessed by the APPLEfast score and prognostic biomarker abundance. Displayed are the 12 biomarkers with the highest degree of correlation with the APPLEfast score. The associated Spearman correlation coefficient ($r_s$) and corresponding unadjusted P-value is displayed inset on each scatterplot.
(TIF)

## Author Contributions

**Conceptualization:** April Summers, Sheng Zhang, Robert Goggs.

**Data curation:** Ruchika Bhawal, Elizabeth T. Anderson, Sheng Zhang, Robert Goggs.

**Formal analysis:** Ruchika Bhawal, Elizabeth T. Anderson, Sheng Zhang, Robert Goggs.

**Funding acquisition:** April Summers, Robert Goggs.

**Investigation:** Brett Montague, April Summers, Ruchika Bhawal, Elizabeth T. Anderson, Sydney Kraus-Malett, Robert Goggs.

**Project administration:** Sydney Kraus-Malett, Robert Goggs.

**Resources:** Sheng Zhang.

**Supervision:** Robert Goggs.

**Validation:** Elizabeth T. Anderson, Sheng Zhang.

**Visualization:** Ruchika Bhawal.

**Writing – original draft:** Brett Montague, Robert Goggs.

**Writing – review & editing:** Brett Montague, April Summers, Ruchika Bhawal, Sheng Zhang, Robert Goggs.

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
