## [Decision Letter · Decision Letter 0]

20 Apr 2022

PONE-D-22-02054Identifying potential biomarkers and therapeutic targets for dogs with sepsis by parallel metabolomics and lipidomics analysesPLOS ONE

Dear Dr. Goggs,

Thank you for submitting your manuscript to PLOS ONE. After careful consideration, we feel that it has merit but does not fully meet PLOS ONE’s publication criteria as it currently stands. Therefore, we invite you to submit a revised version of the manuscript that addresses the points raised during the review process.

We look forward to receiving your revised manuscript.

Kind regards,

Tamil Selvan Anthonymuthu, Ph. D

Academic Editor

PLOS ONE

Journal Requirements:

2. We note that you have referenced (ie. Bewick et al. [5]) which has currently not yet been accepted for publication. Please remove this from your References and amend this to state in the body of your manuscript: (ie “Bewick et al. [Unpublished]”) as detailed online in our guide for authors

Reviewers' comments:

Reviewer's Responses to Questions

**Comments to the Author**

1. Is the manuscript technically sound, and do the data support the conclusions?

Reviewer #1: Yes

2. Has the statistical analysis been performed appropriately and rigorously? 

Reviewer #1: Yes

3. Have the authors made all data underlying the findings in their manuscript fully available?

Reviewer #1: No

4. Is the manuscript presented in an intelligible fashion and written in standard English?

Reviewer #1: Yes

5. Review Comments to the Author

Reviewer #1: The manuscript “Identifying potential biomarkers and therapeutic targets for dogs with sepsis by parallel metabolomics and lipidomics analyses” reports metabolic signatures of sepsis in dogs. The dataset generated from this study will contribute to the canine metabolome database and provide knowledge on septic biomarkers. However, variability within the study populations due to several factors and lack of information on control groups make it unsuitable to publish in its current form. If the authors can provide relevant information and clarify below comments it can be considered for publication in PLosOne.

Major comments:

1. The authors have not mentioned anything about the breed of the dogs used in this study. Given that breed variability in dog plasma metabolome has been reported, this is important to understand how it impacts the data. In addition, there is no information on the breed, age, gender information of the control group.

2. Line 267, Table 1 is confusing with the same variables represented by two different units (SI and US). Is there any specific reasons for showing the same data in two formats? It will more informative to provide comparative values from the control group along with the sepsis group with either US or SI units (if needed the other unit can be given as supplementary data).

3. The title reads “….by parallel metabolomics and lipidomics analyses”. However, the authors have used separate sample extraction methods for small molecule metabolomics and lipidomics instead of using a bi-phasic extraction and subjecting respective phases to metabolomics and lipidomics parallelly. Why? please comment and change title.

4. In the methods section, internal standards were added (after extraction) during resuspension of dried extract for untargeted metabolomics while for untargeted lipidomics internal standards were added during the extraction step. As the latter is the recommended method, why this difference in sample preparation?

5. The authors have acknowledged the diverse diets of sepsis dogs used in this study but have not mentioned the diet of control dogs. As diet influences the metabolome significantly, to make sense of the comparison with control group, this information and any correlation to it is important. Please report information of control group diet. Also, wondering if there was a fasting time (for a short period) for animals prior to sample collection.

6. It will be interesting to check for any correlations between the identified putative metabolic biomarkers and sepsis causes/severity.

7. This study discusses dog sepsis metabolome data in context with human sepsis metabolome data. It will add value to include a venn diagram of metabolic biomarkers identified in this study and those from human sepsis metabolome studies. It will give a visual perspective of unique and common metabolic biomarkers identified in this study.

Minor comments:

1. Line 276: OPLS-DA is mentioned in the test while PCA is mentioned in the figure legend 2 A. Please clarify. Assignment of figure number is not similar in the text (2A is OPLS-DA) and figure legend (2A is volcano plot).

2. Please provide high resolution images.

6. PLOS authors have the option to publish the peer review history of their article (what does this mean?). If published, this will include your full peer review and any attached files.

Reviewer #1: No

---

## [Author Response · Author response to Decision Letter 0]

10 May 2022

For full responses including the additional figure and revised table, please see the attached file "Author replies". A text only summary of our responses is below.

The manuscript “Identifying potential biomarkers and therapeutic targets for dogs with sepsis by parallel metabolomics and lipidomics analyses” reports metabolic signatures of sepsis in dogs. The dataset generated from this study will contribute to the canine metabolome database and provide knowledge on septic biomarkers. However, variability within the study populations due to several factors and lack of information on control groups make it unsuitable to publish in its current form. If the authors can provide relevant information and clarify below comments it can be considered for publication in PLosOne.

>>We thank the reviewer for this feedback. We have added information on the control population and have attempted to address all the concerns raised below.

Major comments:

1. The authors have not mentioned anything about the breed of the dogs used in this study. Given that breed variability in dog plasma metabolome has been reported, this is important to understand how it impacts the data. In addition, there is no information on the breed, age, gender information of the control group.

>>We apologize for this omission. We now include information about the breeds of the dogs with sepsis and the breed, age and sex information for the healthy control dogs, as follows:

“The 20 healthy control dogs consisted of 12 mixed breed dogs and 8 purebred dogs including 2 mastiffs and 1 dog of each of the following breeds: akita, Bernese Mountain dog, golden retriever, Labrador retriever, redbone coonhound, rottweiler. The healthy dogs consisted of 13 spayed females, 6 castrated males and 1 intact male dog. In the healthy dog population, 19/20 dogs were eating commercial cooked diets, with 3 dogs eating more than one type of commercial food, and 1 dog was fed a home-prepared raw food diet. Two healthy dogs were fed supplements intended to maintain joint health including glucosamine and chondroitin, green lipped mussel extract, methyl sulfonyl methane, hyaluronic acid, ascorbic acid, and DL-phenylalanine.”

2. Line 267, Table 1 is confusing with the same variables represented by two different units (SI and US). Is there any specific reasons for showing the same data in two formats? It will more informative to provide comparative values from the control group along with the sepsis group with either US or SI units (if needed the other unit can be given as supplementary data).

>>We have removed the US units and now provide the SI units only. We have also included all the available information about the healthy control dogs. For some of the variables e.g., blood pressure readings, we do not have these values to report for the healthy controls because they were not recorded during the routine physical examinations performed to screen these dogs.

3. The title reads “….by parallel metabolomics and lipidomics analyses”. However, the authors have used separate sample extraction methods for small molecule metabolomics and lipidomics instead of using a bi-phasic extraction and subjecting respective phases to metabolomics and lipidomics parallelly. Why? please comment and change title.

>>The reviewer raises an important question regarding use of a biphasic extraction and subjecting respective phases to metabolomic and lipidomic analysis. In this study, we were not sample limited, and we wished to perform both C18 (reverse phase chromatography) and HILIC (hydrophilic chromatography) with the metabolomic samples. We therefore extracted all types of metabolites after protein precipitation using ice cold 100% methanol and the supernatants were used for further analysis. Whereas, for lipidomics, we extracted the lipids after adding internal standards for 7 different lipid classes to the samples before protein precipitation and liquid-liquid extraction with dichloromethane/methanol and water. This was performed to maximize lipid recovery because higher amounts of organic solvent improved lipid extraction compared with using only methanol as for metabolomics.

In the title, “parallel” term refers to metabolomics and lipidomics analyses done at the same time on the same plasma samples for both healthy dogs and dogs with sepsis. It was not our intention to indicate that we used the same extracted samples for both metabolomics and lipidomics analyses. We recognize that this might be confusing, however, and have therefore removed the word “parallel” from the title of the revised manuscript.

4. In the methods section, internal standards were added (after extraction) during resuspension of dried extract for untargeted metabolomics while for untargeted lipidomics internal standards were added during the extraction step. As the latter is the recommended method, why this difference in sample preparation?

>>We apologize for any confusion we created here. This difference in our use of internal standards reflects fundamental differences between untargeted metabolomics and lipidomics. Lipidomics covers multiple specific classes of lipids and the ionization of different classes of lipids varies. Deuterated labeled lipids of seven different lipid classes are available and allowed us to perform normalization of those different lipid classes throughout all the samples. This enabled us to obtain accurate quantitation of all lipids identified within each class of lipids.

In contrast, in our metabolomic analyses, it wasn’t feasible to include internal standards that cover all metabolite types or classes that could be used for normalization or any other quantitation factor. Rather, we used these internal standards as a quality assurance measure to monitor instrument performance over time and to confirm that injection of each sample is within 5% variation. For our metabolomics, we therefore used global quality control samples (pooled samples of all the samples) to perform normalization of all samples and correct possible performance changes of the mass spectrometer across the data acquisition period.

5. The authors have acknowledged the diverse diets of sepsis dogs used in this study but have not mentioned the diet of control dogs. As diet influences the metabolome significantly, to make sense of the comparison with control group, this information and any correlation to it is important. Please report information of control group diet. Also, wondering if there was a fasting time (for a short period) for animals prior to sample collection.

>>We have included the dietary information for the control dogs. None of the dogs in this study were deliberately fasted prior to blood sampling. Some of the dogs with sepsis were inappetent for variable amounts of time prior to sampling, however. We have added this information to the results and now discuss this as a potential limitation in the manuscript.

Results section:

“None of the dogs sampled for this study were intentionally fasted prior to sample collection. Some dogs with sepsis were inappetent for variable amounts of time prior to sampling, however.”

Discussion section:

“Similarly, we were not able to standardize the diets of the healthy control animals sampled and this likely also introduced variation in our data that is unrelated to sepsis.”

6. It will be interesting to check for any correlations between the identified putative metabolic biomarkers and sepsis causes/severity.

>>The reviewer makes a good suggestion here and we agreed this would be of interest. We calculated non-parametric correlation coefficients (Spearman’s r) between biomarker abundance (after log transformation) and illness severity score (APPLEfast) for the top 50 biomarkers identified in C18 and HILIC analyses. Numerous significant correlations were identified, suggesting that illness severity may underpin the associations between outcome and these biomarkers. Since these analyses are exploratory, we have included them as supplemental material (Figure S1). We have added text to the materials and methods section describing the procedures and the following to the results section:

“To further assess the utility of these biomarkers, non-parametric correlations between the clinical illness severity score (APPLEfast) and the abundances of the top 50 biomarkers for both C18 and HILIC analyses were calculated (Fig S1). Multiple prognostic biomarkers were significantly associated with illness severity, suggesting that prognostic metabolomic biomarkers identified likely reflect the severity of sepsis in dogs.”

We did not perform assessments of associations between biomarkers and sepsis causes because our overall population was heterogeneous with multiple distinct types and locations of the primary cause of sepsis. As such, comparisons between groups would not have been meaningful given the small numbers of dogs that would have been categorized into each of the various groups.

7. This study discusses dog sepsis metabolome data in context with human sepsis metabolome data. It will add value to include a venn diagram of metabolic biomarkers identified in this study and those from human sepsis metabolome studies. It will give a visual perspective of unique and common metabolic biomarkers identified in this study.

>>We thank the reviewer for this suggestion, and we agree. To generate this analysis, we compared our C18 and HILIC curated data (filtered out based on MS2 spectra confirmation, background noise, plasticizers and drug metabolites) and identified 107 metabolites (mostly amino acids, fatty acids, lipids classes like sphingomyelins, phosphocholines, and TCA metabolites like lactate, citrate, oxoglutarate and others) that were common between dogs and humans:

Human data were taken from: Rogers et al. Crit Care Explor. 2021; 3(8):e0478. doi: 10.1097/CCE.0000000000000478.

We have included this analysis, the additional human reference, and the Venn diagram figure in the revised manuscript as follows:

Comparisons with human sepsis

To contextualize the dog sepsis metabolome data using previously reported data on the metabolome in human sepsis metabolome data we compared our C18 and HILIC data with publicly available data from a study of the metabolomes of 197 critically ill humans with early sepsis [29]. Specifically, we compared combined curated C18 and HILIC data (filtered based on MS2 spectra confirmation and curated to remove known exposome compounds, plasticizers, and drug metabolites) with those from Rogers et al. [29]. We identified 107 metabolites (predominantly amino acids, fatty acids, lipids classes including sphingomyelins and phosphocholines, and TCA metabolites like lactate, citrate, oxoglutarate) that were common between dogs and humans (Fig 3, Data S3).

Fig 3. Venn diagram comparing metabolites in human and canine sepsis. Comparisons between metabolites identified in a study of humans with early sepsis and curated combined data from C18 and HILIC analyses in dogs identified 107 compounds in common, with 496 compounds unique to dogs, and 869 compounds unique to humans.

Minor comments:

1. Line 276: OPLS-DA is mentioned in the test while PCA is mentioned in the figure legend 2 A. Please clarify. Assignment of figure number is not similar in the text (2A is OPLS-DA) and figure legend (2A is volcano plot).

>>Thank you for spotting these errors, we have corrected them.

2. Please provide high resolution images.

>>We have uploaded figure images at 300dpi, and these have been approved by the PLOS One PACE software.

---

## [Editor Report · Decision Letter 1]

24 Jun 2022

Identifying potential biomarkers and therapeutic targets for dogs with sepsis by metabolomics and lipidomics analyses

PONE-D-22-02054R1

Dear Dr. Goggs,

We’re pleased to inform you that your manuscript has been judged scientifically suitable for publication and will be formally accepted for publication once it meets all outstanding technical requirements.

Kind regards,

Tamil Selvan Anthonymuthu, Ph. D

Academic Editor

PLOS ONE
---

## [Editor Report · Acceptance letter]

28 Jun 2022

PONE-D-22-02054R1 

Identifying potential biomarkers and therapeutic targets for dogs with sepsis using metabolomics and lipidomics analyses 

Dear Dr. Goggs:

I'm pleased to inform you that your manuscript has been deemed suitable for publication in PLOS ONE. Congratulations! Your manuscript is now with our production department. 

Kind regards, 

on behalf of

Dr. Tamil Selvan Anthonymuthu 

Academic Editor

PLOS ONE